# Entropy Production in Exactly Solvable Systems

**DOI:** 10.3390/e22111252

**Published:** 2020-11-03

**Authors:** Luca Cocconi, Rosalba Garcia-Millan, Zigan Zhen, Bianca Buturca, Gunnar Pruessner

**Affiliations:** 1Department of Mathematics, Imperial College London, 180 Queen’s Gate, London SW7 2AZ, UK; luca.cocconi14@imperial.ac.uk (L.C.); rg646@cam.ac.uk (R.G.-M.); z.zhen19@imperial.ac.uk (Z.Z.); 2Centre for Complexity Science, Imperial College London, London SW7 2AZ, UK; 3The Francis Crick Institute, 1 Midland Rd, London NW1 1AT, UK; 4DAMTP, Centre for Mathematical Sciences, University of Cambridge, Wilberforce Road, Cambridge CB3 0WA, UK; 5Department of Physics, Imperial College London, Exhibition Road, London SW7 2AZ, UK; bianca.buturca16@imperial.ac.uk

**Keywords:** entropy production, active matter, exact results, stochastic thermodynamics

## Abstract

The rate of entropy production by a stochastic process quantifies how far it is from thermodynamic equilibrium. Equivalently, entropy production captures the degree to which global detailed balance and time-reversal symmetry are broken. Despite abundant references to entropy production in the literature and its many applications in the study of non-equilibrium stochastic particle systems, a comprehensive list of typical examples illustrating the fundamentals of entropy production is lacking. Here, we present a brief, self-contained review of entropy production and calculate it from first principles in a catalogue of exactly solvable setups, encompassing both discrete- and continuous-state Markov processes, as well as single- and multiple-particle systems. The examples covered in this work provide a stepping stone for further studies on entropy production of more complex systems, such as many-particle active matter, as well as a benchmark for the development of alternative mathematical formalisms.

## 1. Introduction

Stochastic thermodynamics has progressively evolved into an essential tool in the study of non-equilibrium systems as it connects the quantities of interest in traditional thermodynamics, such as work, heat and entropy, to the properties of microscopically resolved fluctuating trajectories [1,2,3]. The possibility of equipping stochastic processes with a consistent thermodynamic and information-theoretic interpretation has resulted in a number of fascinating works, with the interface between mathematical physics and the biological sciences proving to be a particularly fertile ground for new insights (e.g., [4,5,6,7,8]). The fact that most of the applications live on the small scale is not surprising, since it is precisely at the microscopic scale that fluctuations start to play a non-negligible role.

The concept of entropy and, more specifically, entropy production has attracted particular interest, as a consequence of the quantitative handle it provides on the distinction between equilibrium systems, passive systems relaxing to equilibrium and genuinely non-equilibrium, ‘active’ systems. While there exist multiple routes to the mathematical formulation of entropy production [9,10,11,12,13,14], the underlying physical picture is consistent: the entropy production associated with an ensemble of stochastic trajectories quantifies the degree of certainty with which we can assert that a particular event originates from a given stochastic process or from its suitably defined conjugate (usually, its time-reverse). When averaged over long times (or over an ensemble), a non-vanishing entropy production signals time-reversal symmetry breaking at the microscopic scale. This implies, at least for Markovian systems, the existence of steady-state probability currents in the state space, which change sign under time-reversal. When a thermodynamically consistent description is available, the average rate of entropy production can be related to the rate of energy or information exchange between the system, the heat bath(s) it is connected to, and any other thermodynamic entity involved in the dynamics, such as a measuring device [15,16,17]. Whilst the rate of energy dissipation is of immediate interest since it captures how ‘costly’ it is to sustain specific dynamics (e.g., the metabolism sustaining the development of an organism [18,19]), entropy production has also been found to relate non-trivially to the efficiency and precision of the corresponding process via uncertainty relations [3,20]. Entropy production along fluctuating trajectories also plays a fundamental role in the formulation of various fluctuation theorems [12].

Given the recent interest in stochastic thermodynamics and entropy production in particular, as well as the increasing number of mathematical techniques implemented for the quantification of the latter, it is essential to have available a few, well-understood reference systems, for which exact results are known. These can play the role of benchmarks for new techniques, while helping neophytes to develop intuition. This is not to say that exact results for more complicated systems are not available, see for example [21], however they are usually limited to small systems and/or require numerical evaluation. In this work, we will present results exclusively in the framework proposed by Gaspard [11], specifically in the form of Equations (4), (Equation 15) and (Equation 16), which we review and contextualise by deriving them via different routes in Section 2. In Section 3, we begin the analysis with processes in discrete state space (Section 3.1, Section 3.2, Section 3.3, Section 3.4, Section 3.5, Section 3.6, Section 3.7 and Section 3.8), and subsequently extend it to the continuous case (Section 3.9, Section 3.10 and Section 3.11). Finally, in Section 3.12 and Section 3.13 we consider processes that involve both discrete and continuous degrees of freedom. Time is taken as a continuous variable throughout.

## 2. Brief Review of Entropy Production

*Entropy production of jump processes.* The concept of time-dependent informational entropy associated with a given ensemble of stochastic processes was first introduced by Shannon [22]. For an arbitrary probability mass function $P_{n}\left(t\right)$ of time *t* over a discrete set of states n∈Ω, the Shannon entropy is defined as
(1)$$S\left(t\right)=-\sum_{n}P_{n}\left(t\right)\ln P_{n}\left(t\right)$$
with the convention henceforth of xlnx=0 for x=0. It quantifies the inherent degree of uncertainty about the state of a process. In the microcanonical ensemble $P_{n}$ is constant in *t* and *n* and upon providing an entropy scale in the form of the Boltzmann constant $k_{B}$, Shannon’s entropy reduces to that of traditional thermodynamics given by Boltzmann’s $S=k_{B}\ln\left|\Omega \right|$, where $\left|\Omega \right|=1/P_{n}$ is the cardinality of Ω. In Markovian systems, the probability $P_{n}\left(t\right)$ depends on *n* and evolves in time *t* according to the master equation
(2)$${\dot{P}}_{n}\left(t\right)=\sum_{m}\left(P_{m}\left(t\right)w_{mn}-P_{n}\left(t\right)w_{nm}\right)$$
with non-negative transition rates $w_{mn}$ from state *m* to state n≠m. Equation (Equation 2) reduces to ${\dot{P}}_{n}\left(t\right)=\sum_{m}P_{m}\left(t\right)w_{mn}$ by imposing the Markov condition $\sum_{m}w_{nm}=0$, equivalently $w_{nn}=-\sum_{m\neq n}w_{nm}$, which we will use in the following. For simplicity, we will restrict ourselves to time-independent rates $w_{nm}$ but as far as the following discussion is concerned, generalising to time-dependent rates is a matter of replacing $w_{nm}$ by $w_{nm}\left(t\right)$. The rate of change of entropy for a continuous time jump process can be derived by differentiating S(t) in Equation (Equation 1) with respect to time and substituting (Equation 2) into the resulting expression [11,23], thus obtaining
(3)$$\dot{S}\left(t\right)=-\sum_{m,n}P_{m}\left(t\right)w_{mn}\ln\left(P_{n}\left(t\right)\right)=\sum_{m,n}P_{n}\left(t\right)w_{nm}\ln\left(\frac{P_{n}\left(t\right)}{P_{m}\left(t\right)}\right)={\dot{S}}_{e}\left(t\right)+{\dot{S}}_{i}\left(t\right)$$
where we define
(4a)$$\begin{array}{cc} {\dot{S}}_{e}\left(t\right)\; & =-\frac{1}{2}\sum_{m,n}\left(P_{n}\left(t\right)w_{nm}-P_{m}\left(t\right)w_{mn}\right)\ln\left(\frac{w_{nm}}{w_{mn}}\right)\\ \; & =-\sum_{m,n}P_{n}\left(t\right)w_{nm}\ln\left(\frac{w_{nm}}{w_{mn}}\right)=-\sum_{m,n}\left(P_{n}\left(t\right)w_{nm}-P_{m}\left(t\right)w_{mn}\right)\ln\left(\frac{w_{nm}}{w_{0}}\right) \end{array}$$
(4b)$$\begin{array}{cc} {\dot{S}}_{i}\left(t\right)\; & =\frac{1}{2}\sum_{m,n}\left(P_{n}\left(t\right)w_{nm}-P_{m}\left(t\right)w_{mn}\right)\ln\left(\frac{P_{n}\left(t\right)w_{nm}}{P_{m}\left(t\right)w_{mn}}\right)\\ \; & =\sum_{m,n}P_{n}\left(t\right)w_{nm}\ln\left(\frac{P_{n}\left(t\right)w_{nm}}{P_{m}\left(t\right)w_{mn}}\right)=\sum_{m,n}\left(P_{n}\left(t\right)w_{nm}-P_{m}\left(t\right)w_{mn}\right)\ln\left(\frac{P_{n}\left(t\right)w_{nm}}{w_{0}}\right) \end{array}$$
with arbitrary positive rate $w_{0}$ to restore dimensional consistency, that cancel trivially. Here we follow the convention [1] to split the rate of entropy change into two contributions: the first, Equation ([Disp-formula FD4a-entropy-22-01252]), commonly referred to as “external” entropy production or entropy flow, is denoted by ${\dot{S}}_{e}$. It contains a factor $\ln(w_{nm}/w_{mn})$ corresponding, for systems satisfying local detailed balance, to the net change in entropy of the reservoir(s) associated with the system’s transition from state *n* to state *m*. For such thermal systems, ${\dot{S}}_{e}$ can thus be identified as the rate of entropy production in the environment [9,24]. The second contribution, Equation ([Disp-formula FD4b-entropy-22-01252]), termed “internal” entropy production and denoted by ${\dot{S}}_{i}$ is non-negative because (x−y)ln(x/y)≥0 for any two real, positive *x*, *y* and using the convention zlnz=0 for z=0. The internal entropy production vanishes when the global detailed balance condition $P_{n}\left(t\right)w_{nm}=P_{m}\left(t\right)w_{mn}$ is satisfied for all pairs of states. In this sense, a non-vanishing ${\dot{S}}_{i}$ is the fingerprint of non-equilibrium phenomena. Defining $\pi_{n}=\lim_{t\to \infty }P_{n}\left(t\right)$ as the probability mass function at steady-state, the internal entropy production rate can be further decomposed into two non-negative contributions, ${\dot{S}}_{i}\left(t\right)={\dot{S}}_{i}^{a}\left(t\right)+{\dot{S}}_{i}^{\mathrm{na}}\left(t\right)$, of the form
(5a)$$\begin{array}{cc} {\dot{S}}_{i}^{a}\left(t\right) & =\sum_{m,n}P_{n}\left(t\right)w_{nm}\ln\left(\frac{\pi_{n}w_{nm}}{\pi_{m}w_{mn}}\right) \end{array}$$
(5b)$$\begin{array}{cc} {\dot{S}}_{i}^{\mathrm{na}}\left(t\right) & =\sum_{m,n}P_{n}\left(t\right)w_{nm}\ln\left(\frac{P_{n}\left(t\right)\pi_{m}}{P_{m}\left(t\right)\pi_{n}}\right)=-\sum_{n}{\dot{P}}_{n}\left(t\right)\ln\left(\frac{P_{n}\left(t\right)}{\pi_{n}}\right)\;. \end{array}$$
These contributions are usually referred to as adiabatic (or housekeeping) and non-adiabatic (or excess) entropy production rates, respectively [1]. The non-adiabatic contribution vanishes at steady-state, $\lim_{t\to \infty }{\dot{S}}_{i}^{\mathrm{na}}\left(t\right)=0$. While these quantities have received attention in the context of fluctuation theorems [23], they will not be discussed further here. At steady-state, namely when ${\dot{P}}_{n}\left(t\right)=0$ for all *n*, $\dot{S}\left(t\right)$ in Equation (Equation 3) vanishes by construction, so that the internal and external contributions to the entropy production cancel each other exactly, $\dot{S}\left(t\right)={\dot{S}}_{e}\left(t\right)+{\dot{S}}_{i}\left(t\right)=0$, while they vanish individually only for systems at equilibrium. Equation (4) will be used throughout the present work to compute the entropy productions of discrete-state processes.

*Entropy production as a measure of time-reversal-symmetry breaking.* As it turns out, a deeper connection between internal entropy production and time-reversal symmetry breaking can be established [11]. The result, which we re-derive below, identifies ${\dot{S}}_{i}$ as the relative dynamical entropy (i.e., the Kullback–Leibler divergence [25]) per unit time of the ensemble of forward paths and their time-reversed counterparts. To see this, we first need to define a path $n=(n_{0},n_{1},\ldots ,n_{M})$ as a set of trajectories starting at time $t_{0}$ and visiting states $n_{j}$ at successive discrete times $t_{j}=t_{0}+j\tau$ with j=0,1,…,M, equally spaced by a time interval τ. For a time-homogeneous Markovian jump process in continuous time, the joint probability of observing a particular path is
(6)$$P\left(n;t_{0},M\tau \right)=P_{n_{0}}\left(t_{0}\right)W\left(n_{0}\to n_{1};\tau \right)W\left(n_{1}\to n_{2};\tau \right)\ldots W\left(n_{M-1}\to n_{M};\tau \right)$$
where $P_{n_{0}}\left(t_{0}\right)$ is the probability of observing the system in state $n_{0}$ at time $t_{0}$, while $W(n_{j}\to n_{j+1};\tau )$ is the probability that the system is in state $n_{j+1}$ time τ after being in state $n_{j}$. This probability can be expressed in terms of the transition rate matrix *w* with elements $w_{mn}$. It is $W\left(n\to m;\tau \right)={\left[\exp\left(w\tau \right)\right]}_{nm}$, the matrix elements of the exponential of the matrix wτ with the Markov condition imposed. It can be expanded in small τ as
(7)$$W\left(n\to m;\tau \right)=\delta_{n,m}+w_{nm}\tau +O\left(\tau^{2}\right)\;,$$
where $\delta_{n,m}$ is the Kronecker-δ function. We can now define a dynamical entropy per unit time [22] as
(8)$$h\left(t_{0},\Delta t\right)=\lim_{M\to \infty }-\frac{1}{\Delta t}\sum_{n_{0},\ldots ,n_{M}}P\left(n;t_{0},\Delta t\right)\ln P\left(n;t_{0},\Delta t\right)\;.$$
where the limit is to be considered a continuous time limit taken at fixed $\Delta t=t_{M}-t_{0}=M\tau$ [26], thus determining the sampling interval τ, and the sum runs over all possible paths n. Other than τ, the paths are the only quantity on the right-hand side of Equation (Equation 8) that depend on *M*. The dynamical entropy $h(t_{0},\Delta t)$ may be considered the expectation of $\ln(P\left(n;t_{0},\Delta t\right))$ across all paths. Similarly to the static Shannon entropy, the dynamical entropy $h(t_{0},\Delta t)$ quantifies the inherent degree of uncertainty about the evolution over a time Δt of a process starting at a given time $t_{0}$. To compare with the dynamics as observed under time-reversal, one introduces the time-reversed path $n^{R}=\left(n_{M},n_{M-1},\ldots ,n_{0}\right)$ and thus the time-reversed dynamical entropy per unit time as
(9)$$h^{R}\left(t_{0},\Delta t\right)=\lim_{M\to \infty }-\frac{1}{\Delta t}\sum_{n_{0},\ldots ,n_{M}}P\left(n;t_{0},\Delta t\right)\ln P\left(n^{R};t_{0},\Delta t\right)\;.$$
While similar in spirit to $h(t_{0},\Delta t)$, the physical interpretation of $h^{R}\left(t_{0},\Delta t\right)$ as the expectation of $\ln(P\left(n^{R};\Delta t\right))$ under the forward probability $P(n;t_{0},\Delta t)$ is more convoluted since it involves the forward and the backward paths simultaneously, which have potentially different statistics. However, time-reversal symmetry implies precisely identical statistics of the two ensembles, whence $h\left(t_{0},\Delta t\right)=h^{R}\left(t_{0},\Delta t\right)$. The motivation for introducing $h^{R}\left(t_{0},\Delta t\right)$ is that the difference of the two dynamical entropies defined above is a non-negative Kullback–Leibler divergence given by
(10)$$h^{R}\left(t_{0},\Delta t\right)-h\left(t_{0},\Delta t\right)=\lim_{M\to \infty }\frac{1}{\Delta t}\sum_{n}P\left(n;t_{0},\Delta t\right)\ln\left(\frac{P(n;t_{0},\Delta t)}{P(n^{R};t_{0},\Delta t)}\right)\;.$$
Using Equation (Equation 6) in (Equation 10) with Equation (Equation 7) provides the expansion
(11)$$h^{R}\left(t_{0},\Delta t\right)-h\left(t_{0},\Delta t\right)=\sum_{nm}P_{n}\left(t_{0}\right)w_{nm}\ln\left(\frac{P_{n}\left(t_{0}\right)w_{nm}}{P_{m}\left(t_{0}\right)w_{mn}}\right)+O\left(\Delta t\right)\;,$$
which is an instantaneous measure of the Kullback–Leibler divergence. The limit of $h^{R}\left(t_{0},\Delta t\right)-h\left(t_{0},\Delta t\right)$ in small Δt is finite and identical to the internal entropy production (4b) derived above. This result establishes the profound connection between broken global detailed balance, Equation (4), and Kullback–Leibler divergence, Equation (Equation 11), both of which can thus be recognised as fingerprints of non-equilibrium systems. In light of this connection, it might not come as a surprise that the steady-state rate of entropy production is inversely proportional to the minimal time needed to decide on the direction of the arrow of time [27].

*Entropy production for continuous degrees of freedom.* The results above were obtained for Markov jump processes within a discrete state space. However, the decomposition of the rate of change of entropy in Equation (Equation 3) into internal and external contributions can be readily generalised to Markovian processes with continuous degrees of freedom, for example a spatial coordinate. For simplicity, we will restrict ourselves to processes in one dimension but as far as the following discussion is concerned, generalising to higher dimensions is a matter of replacing spatial derivatives and integrals over the spatial coordinate with their higher dimensional counterparts. The dynamics of such a process with probability density P(x,t) to find it at *x* at time *t* are captured by a Fokker–Planck equation of the form $\dot{P}\left(x,t\right)=-\partial_{x}j\left(x,t\right)$, with *j* the probability current, augmented by an initial condition P(x,0). Starting from the Gibbs–Shannon entropy for a continuous random variable $S\left(t\right)=-\int \;dx\;P\left(x,t\right)\ln(P\left(x,t\right)/P_{0})$ with some arbitrary density scale $P_{0}$ for dimensional consistency, we differentiate with respect to time and substitute $-\partial_{x}j\left(x,t\right)$ for $\dot{P}\left(x,t\right)$ to obtain
(12)$$\dot{S}\left(t\right)=-\int \;dx\;\dot{P}\left(x,t\right)\ln\left(\frac{P(x,t)}{P_{0}}\right)=-\int \;dx\;\frac{(\partial_{x}P\left(x,t\right))j\left(x,t\right)}{P(x,t)}\;,$$
where the second equality follows upon integration by parts using $\int \;dx\;\dot{P}\left(x,t\right)=0$ by normalisation. For the paradigmatic case of an overdamped colloidal particle, which will be discussed in more detail below (Section 3.9, Section 3.10 and Section 3.11), the probability current is given by $j\left(x,t\right)=-D\partial_{x}P\left(x,t\right)+\mu F\left(x,t\right)P\left(x,t\right)$ with local, time-dependent force F(x,t). We can then decompose the entropy production $\dot{S}\left(t\right)={\dot{S}}_{i}\left(t\right)+{\dot{S}}_{e}\left(t\right)$ into internal and external contributions as
(13)$${\dot{S}}_{i}\left(t\right)=\int \;dx\;\frac{j{\left(x,t\right)}^{2}}{DP(x,t)}\geq 0$$
and
(14)$${\dot{S}}_{e}\left(t\right)=-\int \;dx\;\frac{\mu }{D}F\left(x,t\right)j\left(x,t\right)\;,$$
respectively. The Kullback–Leibler divergence between the densities of forward and time-reversed paths can be calculated as outlined above for discrete state systems, thus producing an alternative expression for the internal entropy production in the form
(15)$$\begin{array}{cc} {\dot{S}}_{i}\left(t\right) & =\lim_{\Delta t\to 0}h^{R}\left(t,\Delta t\right)-h\left(t,\Delta t\right)\\ & =\lim_{\tau \to 0}\frac{1}{2\tau }\int \;dx\;\;dx^{\prime }\;\left(P\left(x,t\right)W\left(x\to x^{\prime },\tau \right)-P\left(x^{\prime },t\right)W\left(x^{\prime }\to x,\tau \right)\right)\ln\frac{P\left(x,t\right)W\left(x\to x^{\prime },\tau \right)}{P\left(x^{\prime },t\right)W\left(x^{\prime }\to x,\tau \right)}\;. \end{array}$$
Here we have introduced the propagator $W(x^{\prime }\to x,\tau )$, the probability density that a system observed in state $x^{\prime }$ will be found at *x* time τ later. In general, here and above, the density $W(x\to x^{\prime },\tau )$ depends on the absolute time *t*, which we have omitted here for better readability. The corresponding expression for the entropy flow is obtained by substituting (Equation 15) into the balance equation ${\dot{S}}_{e}\left(t\right)=\dot{S}\left(t\right)-{\dot{S}}_{i}\left(t\right)$, whence
(16)$${\dot{S}}_{e}\left(t\right)=-\lim_{\tau \to 0}\frac{1}{2\tau }\int \;dx\;\;dx^{\prime }\;\left(P\left(x,t\right)W\left(x\to x^{\prime },\tau \right)-P\left(x^{\prime },t\right)W\left(x^{\prime }\to x,\tau \right)\right)\ln\frac{W(x^{\prime }\to x,\tau )}{W(x\to x^{\prime },\tau )}\;.$$
Since $\lim_{\tau \to 0}W\left(x\to x^{\prime },\tau \right)=\delta \left(x-x^{\prime }\right)$ [28] and $P\left(x,t\right)\delta \left(x-x^{\prime }\right)=P\left(x^{\prime },t\right)\delta \left(x^{\prime }-x\right)$, the factor in front of the logarithm in (Equation 15) and (Equation 16) vanishes in the limit of small τ, $\lim_{\tau \to 0}P\left(x,t\right)W\left(x\to x^{\prime };\tau \right)-P\left(x^{\prime },t\right)W\left(x^{\prime }\to x;\tau \right)=0$. Together with the prefactor 1/τ, this necessitates the use of L’Hôpital’s rule
(17)$$\lim_{\tau \to 0}\frac{1}{\tau }\left(P\left(x,t\right)W\left(x\to x^{\prime };\tau \right)-P\left(x^{\prime },t\right)W\left(x^{\prime }\to x;\tau \right)\right)=P\left(x,t\right)\dot{W}\left(x\to x^{\prime }\right)-P\left(x^{\prime },t\right)\dot{W}\left(x^{\prime }\to x\right)$$
where we used the shorthand
(18)$$\dot{W}\left(x\to x^{\prime }\right):=\lim_{\tau \to 0}\frac{d}{d\tau }W\left(x\to x^{\prime };\tau \right)\;,$$
which is generally given by the Fokker–Planck equation of the process, so that
(19)$$\dot{P}\left(x,t\right)=\int \;dx^{\prime }\;P\left(x^{\prime },t\right)\dot{W}\left(x^{\prime }\to x\right)\;.$$
In the continuum processes considered below, in particular Section 3.11, Section 3.12 and Section 3.13, $\dot{W}\left(x\to x^{\prime }\right)$ is a kernel in the form of Dirac δ-functions and derivatives thereof, acting under the integral as the adjoint Fokker–Planck operator on P(x,t). With Equation (Equation 17) the internal entropy production of a continuous process (Equation 15) may conveniently be written as
(20a)$$\begin{array}{cc} {\dot{S}}_{i}\left(t\right) & =\frac{1}{2}\int \;dx\;\;dx^{\prime }\;\left(P\left(x,t\right)\dot{W}\left(x\to x^{\prime }\right)-P\left(x^{\prime },t\right)\dot{W}\left(x^{\prime }\to x\right)\right)\times \lim_{\tau \to 0}\ln\left(\frac{P\left(x,t\right)W\left(x\to x^{\prime };\tau \right)}{P\left(x^{\prime },t\right)W\left(x^{\prime }\to x;\tau \right)}\right) \end{array}$$
(20b)$$\begin{array}{cc} \; & =\int \;dx\;\;dx^{\prime }\;P\left(x,t\right)\dot{W}\left(x\to x^{\prime }\right)\times \lim_{\tau \to 0}\ln\left(\frac{P\left(x,t\right)W\left(x\to x^{\prime };\tau \right)}{P\left(x^{\prime },t\right)W\left(x^{\prime }\to x;\tau \right)}\right) \end{array}$$
(20c)$$\begin{array}{cc} \; & =\int \;dx\;\;dx^{\prime }\;\left(P\left(x,t\right)\dot{W}\left(x\to x^{\prime }\right)-P\left(x^{\prime },t\right)\dot{W}\left(x^{\prime }\to x\right)\right)\times \lim_{\tau \to 0}\ln\left(\frac{P\left(x,t\right)W\left(x\to x^{\prime };\tau \right)}{W_{0}P_{0}}\right) \end{array}$$
with suitable constants $W_{0}$ and $P_{0}$. Correspondingly, the (external) entropy flow (Equation 16) is
(21a)$$\begin{array}{cc} {\dot{S}}_{e}\left(t\right) & =-\frac{1}{2}\int \;dx\;\;dx^{\prime }\;\left(P\left(x,t\right)\dot{W}\left(x\to x^{\prime }\right)-P\left(x^{\prime },t\right)\dot{W}\left(x^{\prime }\to x\right)\right)\times \lim_{\tau \to 0}\ln\left(\frac{W(x\to x^{\prime };\tau )}{W(x^{\prime }\to x;\tau )}\right) \end{array}$$
(21b)$$\begin{array}{cc} \; & =-\int \;dx\;\;dx^{\prime }\;P\left(x,t\right)\dot{W}\left(x\to x^{\prime }\right)\times \lim_{\tau \to 0}\ln\left(\frac{W(x\to x^{\prime };\tau )}{W(x^{\prime }\to x;\tau )}\right) \end{array}$$
(21c)$$\begin{array}{cc} \; & =-\int \;dx\;\;dx^{\prime }\;\left(P\left(x,t\right)\dot{W}\left(x\to x^{\prime }\right)-P\left(x^{\prime },t\right)\dot{W}\left(x^{\prime }\to x\right)\right)\times \lim_{\tau \to 0}\ln\left(\frac{W(x\to x^{\prime };\tau )}{W_{0}}\right)\;. \end{array}$$
All of these expressions assume that the limits of the logarithms exist. Naively replacing them by $\ln(\delta \left(x-x^{\prime }\right)/\delta \left(x^{\prime }-x\right))$ produces a meaningless expression with a Dirac δ-function in the denominator. Equations (20) and (21) are identically obtained in the same manner as Equation (4) with the master Equation (Equation 2) replaced by the Fokker–Planck Equation (Equation 19). All of these expressions, Equations (4), (20) and (21), may thus be seen as Gaspard’s [11] framework.

*Langevin description and stochastic entropy.* We have seen in Equations (Equation 13) and (Equation 14) how the notion of entropy production can be extended to continuous degrees of freedom by means of a Fokker–Planck description of the stochastic dynamics. The Fokker–Plank equation is a deterministic equation for the probability density and thus provides a description at the level of ensembles, rather than single fluctuating trajectories. A complementary description can be provided by means of a Langevin equation of motion, which is instead a stochastic differential equation for the continuous degree of freedom [29]. The presence of an explicit noise term, which usually represents faster degrees of freedom or fluctuations induced by the contact with a heat reservoir, allows for a clearer thermodynamic interpretation. A paradigmatic example is that of the overdamped colloidal particle mentioned above, whose dynamics is described by
(22)$$\dot{x}\left(t\right)=\mu F\left(x,t\right)+\zeta \left(t\right)$$
with μ a mobility, F(x,t) a generic force and ζ(t) a white noise term with covariance $\langle \zeta \left(t\right)\zeta \left(t^{\prime }\right)\rangle =2D\delta \left(t-t^{\prime }\right)$. For one-dimensional motion on the real line, the force F(x,t) can always be written as the gradient of a potential V(x,t), namely $F\left(x,t\right)=-\partial_{x}V\left(x,t\right)$, so that it is conservative. For time-independent, stable potentials, V(x,t)=V(x), this leads over long timeframes to an equilibrium steady-state. This property does not hold in higher dimensions and for different boundary conditions (e.g., periodic), in which case the force F(x,t) need not have a corresponding potential V(x,t) for which F(x,t)=−∇V(x,t) [30].

The concept of entropy is traditionally introduced at the level of ensembles. However, due to its role in fluctuation theorems [1,24], a consistent definition at the level of single trajectories is required. This can be constructed along the lines of [12] by positing the trajectory-dependent entropy $S(x^{*}\left(t\right),t)$ where $x^{*}\left(t\right)$ is a random trajectory as given by Equation (Equation 22) and
(23)$$S\left(x,t\right)=-\ln(P\left(x,t\right)/P_{0})\;.$$
Here, P(x,t) denotes the probability density of finding a particle at position *x* at time *t* as introduced above and $P_{0}$ is a scale as used above to maintain dimensional consistency. Given that $x^{*}\left(t\right)$ is a random variable, so is $S(x^{*}\left(t\right),t)$, which may be regarded as an instantaneous entropy. Taking the total derivative with respect to *t* produces
(24)$$\begin{array}{cc} \frac{d}{dt}S\left(x^{*}\left(t\right),t\right) & =-{\left.\frac{\partial_{t}P\left(x,t\right)}{P(x,t)}\right|}_{x=x^{*}\left(t\right)}-{\left.\frac{\partial_{x}P\left(x,t\right)}{P(x,t)}\right|}_{x=x^{*}\left(t\right)}\circ {\dot{x}}^{*}\left(t\right)\\ \; & =-{\left.\frac{\partial_{t}P\left(x,t\right)}{P(x,t)}\right|}_{x=x^{*}\left(t\right)}+\frac{j(x^{*}\left(t\right),t)}{DP(x^{*}\left(t\right),t)}\circ {\dot{x}}^{*}\left(t\right)-\frac{\mu }{D}F\left(x^{*}\left(t\right),t\right)\circ {\dot{x}}^{*}\left(t\right) \end{array}$$
where we have used the processes’ Fokker–Planck equation $\partial_{t}P\left(x,t\right)=-\partial_{x}j\left(x,t\right)$ with $j\left(x,t\right)=\mu F\left(x,t\right)P\left(x,t\right)-D\partial_{x}P\left(x,t\right)$. The total time derivative has been taken as a conventional derivative implying the Stratonovich convention indicated by ∘, which will become relevant below. The term in (Equation 24) containing $\partial_{t}P\left(x,t\right)$ accounts for changes in the probability density due to its temporal evolution, such as relaxation to a steady state, and any time-dependent driving protocol. The product $F\left(x^{*}\left(t\right),t\right)\circ {\dot{x}}^{*}\left(t\right)$ can be interpreted as a power expended by the force and in the absence of an internal energy of the particle, dissipated in the medium. With Einstein’s relation defining the temperature of T=D/μ of the medium, the last term may be written as
(25)$${\dot{S}}_{m}\left(t\right)=\frac{F\left(x^{*}\left(t\right),t\right)\circ {\dot{x}}^{*}\left(t\right)}{T}$$
and thus interpreted as the entropy change in the medium. Together with the entropy change of the particle, this gives the total entropy change of particle and medium,
(26)$${\dot{S}}_{\mathrm{tot}}\left(t\right)=\frac{d}{dt}S\left(x^{*}\left(t\right),t\right)+{\dot{S}}_{m}\left(t\right)=-{\left.\frac{\partial_{t}P\left(x,t\right)}{P(x,t)}\right|}_{x=x^{*}\left(t\right)}+\frac{j(x^{*}\left(t\right),t)}{DP(x^{*}\left(t\right),t)}\circ {\dot{x}}^{*}\left(t\right)\;,$$
which is a random variable, as it depends on the position $x^{*}\left(t\right)$. It also draws on P(x,t) and j(x,t) which are properties of the ensemble. To make the connection to the entropies constructed above we need to take an ensemble average of the instantaneous ${\dot{S}}_{\mathrm{tot}}\left(t\right)$. To do so, we need an interpretation of the last term of (Equation 26), where the noise ζ(t) of ${\dot{x}}^{*}\left(t\right)$, Equation (Equation 22), multiplies $j\left(x^{*}\left(t\right),t\right)/P\left(x^{*}\left(t\right),t\right)$. Equivalently, we need the joint density $P(x,\dot{x};t)$ of position *x* and velocity $\dot{x}$ at time *t*. In the spirit of Ito, this density trivially factorises into a normally distributed $\dot{x}-\mu F\left(x,t\right)$ and P(x,t) as the increment $\dot{x}dt$ on the basis of (Equation 22) depends only on the particle’s current position x(t). However, this is not so in the Stratonovich interpretation of $P(x,\dot{x};t)$, as here the increment depends equally on x(t) and x(t+dt) [1,31,32]. Taking the ensemble average of ${\dot{S}}_{\mathrm{tot}}$ thus produces
(27)$$\begin{array}{cc} \langle {\dot{S}}_{\mathrm{tot}}\left(t\right)\rangle & =\int \;dx^{*}\;\;d{\dot{x}}^{*}\;{\dot{S}}_{\mathrm{tot}}\left(t\right)P\left(x^{*},{\dot{x}}^{*};t\right)\\ \; & =-\int \;dx^{*}\;\frac{\partial_{t}P\left(x^{*},t\right)}{P(x^{*},t)}\int \;d{\dot{x}}^{*}\;P\left(x^{*},{\dot{x}}^{*};t\right)+\int \;dx^{*}\;\;d{\dot{x}}^{*}\;\frac{j(x^{*},t)}{DP(x^{*},t)}{\dot{x}}^{*}P\left(x^{*},{\dot{x}}^{*};t\right)\;, \end{array}$$
where $x^{*}$ and ${\dot{x}}^{*}$ are now dummy variables. The first term on the right hand side vanishes, because $P\left(x^{*},t\right)=\int \;d{\dot{x}}^{*}\;P\left(x^{*},{\dot{x}}^{*};t\right)$ is the marginal of $P(x^{*},{\dot{x}}^{*};t)$ and $\int \;dx^{*}\;\partial_{t}P\left(x^{*},t\right)=0$ by normalisation. The integral over ${\dot{x}}^{*}$ in the second term produces the expected particle velocity conditional to its position,
(28)$$\langle {\dot{x}}^{*}|x^{*},t\rangle =\int \;d{\dot{x}}^{*}\;{\dot{x}}^{*}\frac{P(x^{*},{\dot{x}}^{*};t)}{P(x^{*},t)}$$
in the Stratonovich sense, where it gives rise to the current [12], $\langle {\dot{x}}^{*}|x^{*},t\rangle =j\left(x^{*},t\right)/P\left(x^{*},t\right)$, so that
(29)$$\langle {\dot{S}}_{\mathrm{tot}}\left(t\right)\rangle =\int \;dx^{*}\;\frac{j^{2}\left(x^{*},t\right)}{DP(x^{*},t)}\geq 0\;,$$
which vanishes only in the absence of any probability current, i.e., in thermodynamic equilibrium. In the Ito sense, the conditional expectation (Equation 28) would have instead given rise to the ensemble-independent drift, $\langle {\dot{x}}^{*}|x^{*},t\rangle =\mu F\left(x^{*},t\right)$. Comparing to Equation (Equation 13), the expectation $\langle {\dot{S}}_{\mathrm{tot}}\left(t\right)\rangle$ turns out to be the internal entropy production ${\dot{S}}_{i}\left(t\right)$, so that ${\dot{S}}_{\mathrm{tot}}\left(t\right)$ of Equation (Equation 26) may be regarded as its instantaneous counterpart.

*Path integral methods.* An interesting aspect of working with the Langevin description is the possibility of casting probability densities p([x];t) for paths $x(t^{\prime })$ with $t^{\prime }\in \left[0,t\right]$ into path integrals, for example in the Onsager–Machlup formalism [33,34]. For the colloidal particle introduced in (Equation 22), it gives $p([x];t)=N\exp\left(-A([x];t)\right)$ with the action functional
(30)$$A\left(\left[x\right];t\right)=\int_{0}^{t}\;dt^{\prime }\;\frac{{\left(\dot{x}\left(t^{\prime }\right)-\mu F\left(x\left(t^{\prime }\right),t^{\prime }\right)\right)}^{\circ 2}}{4D}-\frac{\mu }{2}\int_{0}^{t}\;dt^{\prime }\;\partial_{x}F\left(x\left(t^{\prime }\right),t^{\prime }\right)$$
in the Stratonovich discretisation, which differs from the Ito form only by the second term ([34], Section 4.5), which is the Jacobian of the transform of the noise ζ(t) to x(t), Equation (Equation 22). The Stratonovich form is needed so that the action does not give preference to a particular time direction [35]. This choice plays a role in every product of white noise, as is implicit to $\dot{x}$, and a random variable. We therefore indicate the choice by a ∘ also in powers, reminding us that $F(x\left(t^{\prime }\right),t^{\prime })$ should be read as $F(\left(x\left(t^{\prime }\right)+x\left(t^{\prime }+\Delta t\right)\right)/2,t^{\prime }+\Delta t)$ and $\dot{x}\left(t^{\prime }\right)$ as $(x\left(t^{\prime }+\Delta t\right)-x\left(t^{\prime }\right))/2$ with discretisation time step Δt. Evaluating the action for the reversed path $x^{R}\left(t^{\prime }\right)=x\left(t-t^{\prime }\right)$ then gives
(31)$$\begin{array}{cc} A(\left[x^{R}\right];t) & =\int_{0}^{t}\;dt^{\prime }\;\frac{{\left({\dot{x}}^{R}\left(t^{\prime }\right)-\mu F\left(x^{R}\left(t^{\prime }\right),t^{\prime }\right)\right)}^{\circ 2}}{4D}-\frac{\mu }{2}\int_{0}^{t}\;dt^{\prime }\;\partial_{x}F\left(x^{R}\left(t^{\prime }\right),t^{\prime }\right) \end{array}$$
(32)$$\begin{array}{cc} \; & =\int_{0}^{t}\;dt^{\prime }\;\frac{{\left(\dot{x}\left(t^{\prime }\right)+\mu F\left(x\left(t^{\prime }\right),t-t^{\prime }\right)\right)}^{\circ 2}}{4D}-\frac{\mu }{2}\int_{0}^{t}\;dt^{\prime }\;\partial_{x}F\left(x\left(t^{\prime }\right),t-t^{\prime }\right)\;. \end{array}$$
If the force is even under time reversal, $F\left(x,t^{\prime }\right)=F\left(x,t-t^{\prime }\right)$, in particular when it is independent of time, the path probability density obeys
(33)$$\ln\frac{p([x];t)}{p(\left[x^{R}\right];t)}=\int_{0}^{t}\;dt^{\prime }\;\frac{F\left(x\left(t^{\prime }\right),t^{\prime }\right)\circ \dot{x}\left(t^{\prime }\right)}{T}=S_{m}\left(t\right)\;,$$
with random variables multiplied with Stratonovich convention. With Equation (Equation 25), the integral in Equation (Equation 33) can be identified as the entropy of the medium. When the driving is time-independent and the system’s probability distribution eventually becomes stationary, such that $\lim_{t\to \infty }\langle \dot{S}\left(x^{*},t\right)\rangle =0$, Equation (Equation 23), the only contribution to the total entropy change is due to change of entropy in the medium, Equation (Equation 26). Assuming that the system is ergodic, we have the equivalence $\lim_{t\to \infty }S_{m}\left(t\right)/t=\lim_{t\to \infty }\langle {\dot{S}}_{\mathrm{tot}}\left(t\right)\rangle$, where 〈•〉 denotes an ensemble average. Using Equations (Equation 13) and (Equation 29) gives $\lim_{t\to \infty }S_{m}\left(t\right)/t=\lim_{t\to \infty }{\dot{S}}_{i}\left(t\right)$. Equation (Equation 33) can therefore be used directly to compute the steady-state internal entropy production rate. The equivalence between the long-time limit t→∞ and the ensemble average holds only for ergodic systems, whose unique steady-state does not depend on the specific initialisation x(0). This connection between stochastic thermodynamics and field theory has stimulated a number of works aimed at characterising the non-equilibrium features of continuum models of active matter [13,36]. Extensions of this formalism to systems driven by correlated noise have also been proposed [37].

## 3. Systems

In this section, we calculate the entropy production rate on the basis of Gaspard’s framework [11], Equations (4), (Equation 15) and (Equation 16), for different particle systems. We cover the systems listed in Table 1, with both discrete and continuous states and with one or multiple particles.

### 3.1. Two-State Markov Process

Consider a particle that hops between two states, 1 and 2, with transition rates $\dot{W}\left(1\to 2\right)=\alpha$ and $\dot{W}\left(2\to 1\right)=\beta$, see Figure 1 [23,38], and using the notation in Equation (Equation 18) for discrete states. The rate-matrix (see Equation (Equation 7)) may thus be
(34)$$w=\left(\begin{array}{cc} -\alpha & \alpha \\ \beta & -\beta \end{array}\right)\;,$$
with $P\left(t\right)=(P_{1}\left(t\right),P_{2}\left(t\right))$ the probability of the particle to be in state 1 or 2 respectively as a function of time. By normalisation, $P_{1}\left(t\right)+P_{2}\left(t\right)=1$, with probabilistic initial condition P(0)=(p,1−p). Solving the master equation in Equation (Equation 2) yields
(35)$$P\left(t\right)=\left(P_{1}\left(t\right),P_{2}\left(t\right)\right)=\frac{1}{\alpha +\beta }\left(\begin{array}{c} \beta +r\;e^{-(\alpha +\beta )t},\alpha -r\;e^{-(\alpha +\beta )t} \end{array}\right)\;,$$
with r=αp−β(1−p), corresponding to an exponentially decaying probability current
(36)$$P_{1}\left(t\right)\alpha -P_{2}\left(t\right)\beta =r\;e^{-(\alpha +\beta )t}\;.$$

The internal entropy production (4b) is then
(37)$${\dot{S}}_{i}\left(t\right)=\left[P_{1}\left(t\right)\alpha -P_{2}\left(t\right)\beta \right]\ln\left[\frac{P_{1}\left(t\right)\alpha }{P_{2}\left(t\right)\beta }\right]=re^{-(\alpha +\beta )t}\ln\left[\frac{1+\frac{r}{\beta }\;e^{-(\alpha +\beta )t}}{1-\frac{r}{\alpha }\;e^{-(\alpha +\beta )t}}\right]\;,$$
see Figure 2, and the entropy flow ([Disp-formula FD4a-entropy-22-01252]),
(38)$${\dot{S}}_{e}\left(t\right)=-r\;e^{-(\alpha +\beta )t}\;\ln\left(\frac{\alpha }{\beta }\right)\;.$$
At stationarity, ${\dot{S}}_{i}={\dot{S}}_{e}=0$ and therefore the two-state Markov process reaches equilibrium. In this example, the topology of the transition network does not allow a sustained current between states, which inevitably leads to equilibrium in the steady state and, therefore, there is production of entropy only due to the relaxation of the system from the initial state.

### 3.2. Three-State Markov Process

We extend the system in Section 3.1 to three states, 1, 2 and 3, with transition rates $\dot{W}\left(1\to 2\right)=\alpha$, $\dot{W}\left(2\to 3\right)=\alpha$, $\dot{W}\left(3\to 1\right)=\alpha$, $\dot{W}\left(2\to 1\right)=\beta$, $\dot{W}\left(3\to 2\right)=\beta$, and $\dot{W}\left(1\to 3\right)=\beta$, see Figure 3, and using the notation Equation (Equation 18) for discrete states. The rate matrix (see Equation (Equation 7)) is then
(39)$$w=\left(\begin{array}{ccc} -(\alpha +\beta ) & \alpha & \beta \\ \beta & -(\alpha +\beta ) & \alpha \\ \alpha & \beta & -(\alpha +\beta ) \end{array}\right)\;.$$
Assuming the initial condition P(0)=(1,0,0), the probabilities of states 1, 2 and 3 respectively, evolve according to Equation (Equation 2), which has solution
(40a)$$\begin{array}{cc} P_{1}\left(t\right)= & \frac{1}{3}\left(1+2e^{-3\phi t}\cos\left(\sqrt{3}\psi t\right)\right)\;, \end{array}$$
(40b)$$\begin{array}{cc} P_{2}\left(t\right)= & \frac{1}{3}\left(1-2e^{-3\phi t}\cos\left(\sqrt{3}\psi t-\pi /3\right)\right)\;, \end{array}$$
(40c)$$\begin{array}{cc} P_{3}\left(t\right)= & \frac{1}{3}\left(1-2e^{-3\phi t}\cos\left(\sqrt{3}\psi t+\pi /3\right)\right)\;, \end{array}$$
with ϕ=(α+β)/2 and ψ=(α−β)/2.

The entropy production (4b) is then, using (40),
(41)$$\begin{array}{cc} {\dot{S}}_{i}\left(t\right) & =\left(\alpha -\beta \right)\ln\frac{\alpha }{\beta }+\left(P_{1}\left(t\right)\alpha -P_{2}\left(t\right)\beta \right)\ln\left(\frac{P_{1}\left(t\right)}{P_{2}\left(t\right)}\right)\\ \; & +\left(P_{2}\left(t\right)\alpha -P_{3}\left(t\right)\beta \right)\ln\left(\frac{P_{2}\left(t\right)}{P_{3}\left(t\right)}\right)+\left(P_{3}\left(t\right)\alpha -P_{1}\left(t\right)\beta \right)\ln\left(\frac{P_{3}\left(t\right)}{P_{1}\left(t\right)}\right)\;, \end{array}$$
see Figure 2, and the entropy flow ([Disp-formula FD4a-entropy-22-01252]),
(42)$${\dot{S}}_{e}\left(t\right)=-\left(\alpha -\beta \right)\ln\left(\frac{\alpha }{\beta }\right)\;,$$
which is constant throughout. At stationarity, the system is uniformly distributed and, if α≠β, the entropy production and flow satisfy ${\dot{S}}_{i}=-{\dot{S}}_{e}\neq 0$. If α≠β, the particle has a net drift that sustains a probability current (α−β)/3 in the system, which prevents the system from reaching equilibrium. This setup can be generalised straightforwardly to *M*-state Markov processes characterised by the same cyclic structure $\dot{W}\left(i\to i+1\;\left(\mathrm{mod}\;M\right)\right)=\alpha$ and $\dot{W}\left(i\to i-1\;\left(\mathrm{mod}\;M\right)\right)=\beta$, to find that the steady-state entropy production is independent of *M*. These can be seen as simple models of protein conformational cycles driven by instantaneous energy inputs of order $k_{B}T\ln\left(\alpha /\beta \right)$, for example ATP hydrolysis [39].

### 3.3. Random Walk on a Complete Graph

Consider a random walker on a complete graph with *d* nodes, where each node is connected to all other nodes, and the walker jumps from node j∈{1,2,…,d} to node k∈{1,2,…,d}, k≠j, with rate $w_{jk}$, see Figure 4. These are the off-diagonal elements of the corresponding Markov matrix whose diagonal elements are $w_{jj}=-\sum_{i=1,i\neq j}^{d}w_{ji}$. The probability vector $P\left(t\right)=(P_{1}\left(t\right),P_{2}\left(t\right),\ldots ,P_{d}\left(t\right))$ has components $P_{j}\left(t\right)$ that are the probability that the system is in state *j* at time *t*. The general case of arbitrary transition rates is impossible to discuss exhaustively. In the uniform case, $w_{jk}=\alpha$, the Markov matrix has only two distinct eigenvalues, namely eigenvalue αd with degeneracy d−1 and eigenvalue 0 with degeneracy 1. Assuming an arbitrary initial condition P(0), the probability distribution at a later time *t* is
(43)$$P_{j}\left(t\right)=\frac{1}{d}+e^{-d\alpha t}\left(P_{j}\left(0\right)-\frac{1}{d}\right)\;.$$
The steady state, which is associated with the vanishing eigenvalue, is the uniform distribution $\lim_{t\to \infty }P_{j}\left(t\right)=1/d$ for all j∈{1,2,…,d}. The entropy production (4b) of the initial state relaxing to the uniform state is
(44)$${\dot{S}}_{i}\left(t\right)=\frac{1}{2}\alpha e^{-d\alpha t}\sum_{j,k}\left(P_{j}\left(0\right)-P_{k}\left(0\right)\right)\ln\left(\frac{1+e^{-d\alpha t}\left(P_{j}\left(0\right)d-1\right)}{1+e^{-d\alpha t}\left(P_{k}\left(0\right)d-1\right)}\right)\;,$$
and the entropy flow ([Disp-formula FD4a-entropy-22-01252]) is ${\dot{S}}_{e}=0$ throughout. If the walker is initially located on node *k*, so that $P_{j}\left(0\right)=\delta_{j,k}$, the entropy production simplifies to
(45)$${\dot{S}}_{i}\left(t\right)=\left(d-1\right)\alpha e^{-d\alpha t}\ln\left(1+\frac{de^{-d\alpha t}}{1-e^{-d\alpha t}}\right)\;.$$
We can see that the system reaches equilibrium at stationarity, since $\lim_{t\to \infty }{\dot{S}}_{i}\left(t\right)={\dot{S}}_{e}\left(t\right)=0$. Over long timeframes ($de^{-d\alpha t}\ll 1$), the asymptotic behaviour of ${\dot{S}}_{i}$ is
(46)$${\dot{S}}_{i}\left(t\right)=d\left(d-1\right)\alpha e^{-2d\alpha t}+O\left(e^{-3d\alpha t}\right)\;,$$
by expanding the logarithm in the small exponential.

### 3.4. *N* Independent, Distinguishable Markov Processes

In the following, we consider *N* non-interacting, distinguishable particles undergoing Markovian dynamics on a discrete state space, see Figure 5. Each of the *N* particles carries an index ℓ∈{1,2,…,N} and is in state $n_{\ell }\in \left\{1,2,\ldots ,d_{\ell }\right\}$, so that the state of the entire system is given by an *N*-particle state $n=(n_{1},n_{2},\ldots ,n_{N})$. Particle distinguishability implies the factorisation of state and transition probabilities into their single-particle contributions, whence the joint probability $P_{n}\left(t\right)$ of an *N*-particle state n factorises into a product of single particle probabilities $P_{n_{\ell }}^{\left(\ell \right)}\left(t\right)$ of particle *ℓ* to be in state $n_{\ell }$,
(47)$$P_{n}\left(t\right)=\prod_{\ell =1}^{N}P_{n_{\ell }}^{\left(\ell \right)}\left(t\right)\;.$$
Further, the Poissonian rate $w_{nm}$ from *N*-particle state n to *N*-particle state m≠n vanishes for all transitions n→m that differ in more than one component *ℓ*, i.e., $w_{nm}=0$ unless there exists a single ℓ∈{1,2,…,N} such that $m_{k}=n_{k}$ for all k≠ℓ, in which case $w_{nm}=w_{n_{\ell }m_{\ell }}^{\left(\ell \right)}$, the transition rates of the single particle transition of particle *ℓ*.

The entropy production of this *N*-particle system according to Equation (4b),
(48)$${\dot{S}}_{i}\left(t\right)=\frac{1}{2}\sum_{nm}\left(P_{n}\left(t\right)w_{nm}-P_{m}\left(t\right)w_{mn}\right)\ln\left(\frac{P_{n}\left(t\right)w_{nm}}{P_{m}\left(t\right)w_{mn}}\right)$$
simplifies considerably due to $w_{nm}$, as the sum may be re-written as
(49)$$\sum_{nm}\ldots w_{nm}\ldots =\sum_{n}\sum_{\ell }\sum_{m_{\ell }}\ldots w_{nm_{\ell }}\ldots$$
with $m_{\ell }=\left(n_{1},n_{2},\ldots ,n_{\ell -1},m_{\ell },n_{\ell +1},\ldots ,n_{N}\right)$ so that $w_{nm_{\ell }}=w_{n_{\ell }m_{\ell }}^{\left(\ell \right)}$ and
(50)$${\dot{S}}_{i}\left(t\right)=\frac{1}{2}\sum_{n}\sum_{\ell =1}^{N}\sum_{m_{\ell }}\left\{\left(\prod_{k=1}^{N}P_{n_{k}}^{\left(k\right)}\left(t\right)\right)w_{n_{\ell }m_{\ell }}^{\left(\ell \right)}-\left(\prod_{k=1}^{N}P_{m_{k}}^{\left(k\right)}\left(t\right)\right)w_{m_{\ell }n_{\ell }}^{\left(\ell \right)}\right\}\ln\left(\frac{\prod_{k=1}^{N}P_{n_{k}}^{\left(k\right)}\left(t\right)w_{n_{\ell }m_{\ell }}^{\left(\ell \right)}}{\prod_{k=1}^{N}P_{m_{k}}^{\left(k\right)}\left(t\right)w_{m_{\ell }n_{\ell }}^{\left(\ell \right)}}\right)\;.$$
Since $m_{k}=n_{k}$ for any k≠ℓ inside the curly bracket, we may write
(51)$$\begin{array}{ccccc} \prod_{k=1}^{N}P_{n_{k}}^{\left(k\right)}\left(t\right)=P_{n_{\ell }}^{\left(\ell \right)}\left(t\right)\prod_{\begin{array}{c} k=1\\ k\neq \ell \end{array}}^{N}P_{n_{k}}^{\left(k\right)}\left(t\right)\; & \; & \;\mathrm{and}\; & \; & \prod_{k=1}^{N}P_{m_{k}}^{\left(k\right)}\left(t\right)=P_{m_{\ell }}^{\left(\ell \right)}\left(t\right)\prod_{\begin{array}{c} k=1\\ k\neq \ell \end{array}}^{N}P_{n_{k}}^{\left(k\right)}\left(t\right)\;. \end{array}$$
The product $\prod_{k\neq \ell }^{N}P_{n_{k}}^{\left(k\right)}\left(t\right)$ can thus be taken outside the curly bracket in Equation (Equation 50) and be summed over, as well as cancelled in the logarithm. After changing the dummy variables in the remaining summation from $n_{\ell }$ and $m_{\ell }$ to *n* and *m* respectively, the entropy production is
(52)$${\dot{S}}_{i}\left(t\right)=\frac{1}{2}\sum_{\ell =1}^{N}\sum_{nm}\left(P_{n}^{\left(\ell \right)}\left(t\right)w_{nm}^{\left(\ell \right)}-P_{m}^{\left(\ell \right)}\left(t\right)w_{mn}^{\left(\ell \right)}\right)\ln\left(\frac{P_{n}^{\left(\ell \right)}\left(t\right)w_{nm}^{\left(\ell \right)}}{P_{m}^{\left(\ell \right)}\left(t\right)w_{mn}^{\left(\ell \right)}}\right)\;,$$
which is the sum of the entropy productions of the single particle ℓ∈{1,2,…,N}, Equation (4b), irrespective of how each particle is initialised. The same argument applies to ${\dot{S}}_{e}$, the entropy flow Equation ([Disp-formula FD4a-entropy-22-01252]). The entropy production and flow obviously simplify to an *N*-fold product of the single particle expressions if $w_{nm}^{\left(\ell \right)}$ do not depend on *ℓ* and all particles are initialised by the same $P_{n}^{\ell }\left(0\right)$ independent of *ℓ*. This result may equally be found from the dynamical entropy per unit time, Equation (Equation 8).

### 3.5. N Independent, Indistinguishable Two-State Markov Processes

Suppose that *N* identical, indistinguishable, non-interacting particles follow the two-state Markov process described in Section 3.1, Figure 6 [23]. There are Ω=N+1 distinct states given by the occupation number n∈{0,1,…,N} of one of the two states, say state 1, as the occupation number of the other state follows as N−n given the particle number *N* is fixed under the dynamics. In the following, P(n,t) denotes the probability of finding *n* particles in state 1 at time *t*. The master equation is then
(53)$$\dot{P}\left(n,t\right)=-\alpha nP\left(n,t\right)+\alpha \left(n+1\right)P\left(n+1,t\right)-\beta \left(N-n\right)P\left(n,t\right)+\beta \left(N-n+1\right)P\left(n-1,t\right)\;.$$

The state space and the evolution in it can be thought of as a hopping process on a one-dimensional chain of states with non-uniform rates. Provided P(n,0) initially follows a binomial distribution, P(n,0)=Nnpn(1−p)N−n with probability *p* for a particle to be placed in state 1 initially, the solution of Equation (Equation 53) is easily constructed from the solution $P_{1}\left(t\right)$ in Equation (Equation 35) of Section 3.1 via
(54)P(n,t)=NnP1n(t)(1−P1(t))N−nfor0≤n≤N
with $P_{1}\left(0\right)=p$, as ${\dot{P}}_{1}\left(t\right)=-\alpha P_{1}\left(t\right)+\beta \left(1-P_{1}\left(t\right)\right)$, which can be verified by substituting Equation (Equation 54) into Equation (Equation 53). Using Equations (Equation 34) and (Equation 54) in (4b), the entropy production reads
(55a)$$\begin{array}{cc} {\dot{S}}_{i}\left(t\right) & =\sum_{n=1}^{N}\left[P\left(n,t\right)\alpha n-P\left(n-1,t\right)\beta \left(N-n+1\right)\right]\ln\left[\frac{P(n,t)\alpha n}{P(n-1,t)\beta (N-n+1)}\right] \end{array}$$
(55b)$$\begin{array}{cc} & =N\left[P_{1}\left(t\right)\alpha -\left(1-P_{1}\left(t\right)\right)\beta \right]\ln\left[\frac{P_{1}\left(t\right)\alpha }{(1-P_{1}\left(t\right))\beta }\right]\;, \end{array}$$
which is the *N*-fold multiple of the result of the corresponding single particle system, Equation (Equation 37). This result, Equation ([Disp-formula FD55b-entropy-22-01252]), depends on the initialisation being commensurable with Equation (Equation 54) which otherwise is recovered only asymptotically and only if the stationary distribution is unique.

Further, the entropy production of *N* indistinguishable particles being the *N*-fold entropy production of a single particle does not extend to the external entropy flow, which lacks the simplification of the logarithm and gives
(56)S˙e(t)=−N[αP1(t)−β(1−P1(t))]lnαβ+∑n=0N−1P1n(t)(1−P1(t))N−1−nN−1nlnn+1N−n
thus picking up a correction in the form of the additional sum in the curly bracket that vanishes only at N=1 or $P_{1}\left(t\right)=1/2$, but does not contribute at stationarity because of the overall prefactor $\alpha P_{1}-\beta \left(1-P_{1}\right)$ that converges to 0. To make sense of this correction in relation to particle indistinguishability, with the help of Equation (Equation 54) we can rewrite the difference between the right hand side of Equation (Equation 56) and the *N*-fold entropy flow of a single two-state system (Equation 38) as
(57)−N[αP1(t)−β(1−P1(t))]∑n=0N−1P1n(t)(1−P1(t))N−1−nN−1nlnn+1N−n=−∑n=0N−1[α(n+1)P(n+1,t)−β(N−n)P(n,t)]lnn+1N−n
which now explicitly involves the net probability current from the occupation number state with n+1 particles in state *A* to that with *n* particles in state *A*, as well as a the logarithm
(58)lnn+1N−n=lnNn−lnNn+1.
Written in terms of the same combinatorial factors appearing in Equation (Equation 54), the logarithm (Equation 58) can be interpreted as a difference of microcanonical (Boltzmann) entropies, defined as the logarithm of the degeneracy of the occupation number state if we were to assume that the *N* particles are distinguishable. With the help of the master Equation (Equation 53) as well as Equation (Equation 54) and (Equation 58), the term Equation (Equation 57) may be rewritten to give
(59)S˙e(t)=−N[αP1(t)−β(1−P1(t))]lnαβ−∑n=0N−1P˙(n,t)lnNn
This result is further generalised in Equation (Equation 70).

### 3.6. N Independent, Indistinguishable d-State Processes

We generalise now the results in Section 3.3 and Section 3.5 to *N* independent *d*-state Markov processes, see Figure 7. These results represent a special case of those obtained in [40] when the *N* processes are non-interacting. In this section, we consider non-interacting, indistinguishable particles hopping on a graph of *d* nodes with edge-dependent hopping rates $w_{jk}$. As in the two-state system in Section 3.5, we find that the internal (but not the external) entropy production of the *d*-state system ${\dot{S}}_{i}$ is *N* times the entropy production of the individual processes assuming the initial condition is probabilistically identical for all single-particle sub-systems. The entropy productions of a single such process according to Equation (4) read
(60a)$$\begin{array}{cc} {\dot{S}}_{i}^{\left(1\right)}\left(t\right) & =\frac{1}{2}\sum_{jk}\left[P_{j}\left(t\right)w_{jk}-P_{k}\left(t\right)w_{kj}\right]\ln\left(\frac{P_{j}\left(t\right)w_{jk}}{P_{k}\left(t\right)w_{kj}}\right)\;, \end{array}$$
(60b)$$\begin{array}{cc} {\dot{S}}_{e}^{\left(1\right)}\left(t\right) & =-\frac{1}{2}\sum_{jk}\left[P_{j}\left(t\right)w_{jk}-P_{k}\left(t\right)w_{kj}\right]\ln\left(\frac{w_{jk}}{w_{kj}}\right)\;, \end{array}$$
where $P_{j}\left(t\right)$ is the time-dependent probability of a single-particle process to be in state *j*, Section 3.3.

To calculate the entropy production of the *N* concurrent indistinguishable processes using the occupation number representation, we first derive the probability of an occupation number configuration $n=(n_{1},n_{2},\ldots ,n_{d})$, with $\sum_{j=1}^{d}n_{j}=N$, which similarly to Equation (Equation 54) is given by the multinomial distribution
(61)$$P_{n}\left(t\right)=N!\prod_{j=1}^{d}\frac{P_{j}^{n_{j}}\left(t\right)}{n_{j}!}$$
for the probability $P_{n}\left(t\right)$ of the system to be in state n at time *t* assuming that each particle is subject to the same single-particle distribution $P_{j}\left(t\right)$, j∈{1,2,…,d} for all *t*, i.e., in particular assuming that all particles are initialised identically, by placing them all at the same site or, more generally, by placing them initially according to the same distribution $P_{j}\left(0\right)$. Given this initialisation, Equation (Equation 61) solves Equation (Equation 2)
(62)$${\dot{P}}_{n}\left(t\right)=\sum_{m}P_{m}\left(t\right)w_{mn}-P_{n}\left(t\right)w_{nm}$$
with the transition rates $w_{mn}$ discussed below.

For non-interacting processes with a unique stationary distribution, Equation (Equation 61) is always obeyed in the limit of long times after initialisation, since the single-particle distributions $P_{j}\left(t\right)$ are identical at steady state. The entropy production Equation (4b) of the entire system has the same form as Equation (Equation 48) of Section 3.4 (*N* independent, *distinguishable* particles) with $w_{nm}$, however now the transition rate between the occupation number state $n=(n_{1},n_{2},\ldots ,n_{d})$ with $0\leq n_{k}\leq N$ to occupation number state $m=(m_{1},m_{2},\ldots ,m_{d})$. The rate $w_{nm}$ vanishes except when m differs from n in exactly two distinct components, say $m_{j}=n_{j}-1\geq 0$ and $m_{k}=n_{k}+1\geq 1$ in which case $w_{nm}=n_{j}w_{jk}$ with $w_{jk}$ the transition rates of a single particle from *j* to *k* as introduced above. For such m, the rate obeys $w_{mn}=m_{k}w_{kj}$ and the probability $P_{m}\left(t\right)$ fulfills
(63)$$P_{m}\left(t\right)=P_{n}\left(t\right)\frac{P_{k}\left(t\right)n_{j}}{P_{j}\left(t\right)m_{k}}=P_{n}\left(t\right)\frac{P_{k}\left(t\right)w_{nm}w_{kj}}{P_{j}\left(t\right)w_{mn}w_{jk}}\;,$$
which simplifies the entropy production Equation (Equation 48) to
(64)$$\begin{array}{cc} {\dot{S}}_{i}\left(t\right) & =\frac{1}{2}\sum_{nm}\left(P_{n}\left(t\right)w_{nm}-P_{m}\left(t\right)w_{mn}\right)\ln\left(\frac{P_{n}\left(t\right)w_{nm}}{P_{m}\left(t\right)w_{mn}}\right)\\ \; & =\frac{1}{2}\sum_{n}\sum_{jk}\left(P_{n}\left(t\right)n_{j}w_{jk}-P_{m}\left(t\right)m_{k}w_{kj}\right)\ln\left(\frac{P_{j}\left(t\right)w_{jk}}{P_{k}\left(t\right)w_{kj}}\right) \end{array}$$
where the sum $\sum_{n}$ runs over all allowed configurations, namely $0\leq n_{j}\leq N$ for j=1,2,…,d with $\sum_{j}n_{j}=N$ and $m=(n_{1},n_{2},\ldots ,n_{j}-1,\ldots ,n_{k}+1,\ldots ,n_{d})$ is derived from n as outlined above. Strictly, $P_{n}\left(t\right)$ has to be defined to vanish for invalid states n, so that the first bracket in the summand of Equation (Equation 64) vanishes, in particular when $n_{j}=0$, in which case $m_{j}=-1$. To proceed, we introduce the probability
(65)$${\bar{P}}_{{\bar{n}}_{j}}\left(t\right)=\left(N-1\right)!\frac{P_{j}^{n_{j}-1}}{(n_{j}-1)!}\prod_{i=1,i\neq j}^{d}\frac{P_{i}^{n_{i}}}{n_{i}!}\;,$$
defined to vanish for $n_{j}=0$, so that $P_{n}\left(t\right)n_{j}=NP_{j}\left(t\right){\bar{P}}_{{\bar{n}}_{j}}\left(t\right)$. The probability ${\bar{P}}_{{\bar{n}}_{j}}\left(t\right)$ is that of finding $n_{i}$ particles at states i≠j and $n_{j}-1$ particles at state *j*. It is Equation (Equation 61) evaluated in a system with only N−1 particles and configuration ${\bar{n}}_{j}=\left(n_{1},n_{2},\ldots ,n_{j-1},n_{j}-1,n_{j+1},\ldots ,n_{d}\right)={\bar{m}}_{k}$ a function of n. Equation (Equation 64) may now be rewritten as
(66)$${\dot{S}}_{i}\left(t\right)=\frac{N}{2}\sum_{jk}\left\{\left(\sum_{n}{\bar{P}}_{{\bar{n}}_{j}}\left(t\right)\right)P_{j}\left(t\right)w_{jk}-\left(\sum_{n}{\bar{P}}_{{\bar{m}}_{k}}\left(t\right)\right)P_{k}\left(t\right)w_{kj}\right\}\ln\left(\frac{P_{j}\left(t\right)w_{jk}}{P_{k}\left(t\right)w_{kj}}\right)$$
where we have used that the arguments of the logarithm are independent of n and m. The summation over n gives
(67)$$\sum_{n}{\bar{P}}_{{\bar{n}}_{j}}\left(t\right)=\sum_{n}{\bar{P}}_{{\bar{m}}_{k}}\left(t\right)=1$$
so that
(68)$${\dot{S}}_{i}\left(t\right)=\frac{N}{2}\sum_{jk}\left(P_{j}\left(t\right)w_{jk}-P_{k}\left(t\right)w_{kj}\right)\ln\left(\frac{P_{j}\left(t\right)w_{jk}}{P_{k}\left(t\right)w_{kj}}\right)=N{\dot{S}}_{i}^{\left(1\right)}\left(t\right)$$
which is the *N*-fold entropy production of the single particle system ${\dot{S}}_{i}\left(t\right)$, Equation ([Disp-formula FD60a-entropy-22-01252]), or equivalently that of *N* distinguishable particles, Equation (Equation 52), Section 3.4. As in Section 3.5, this dramatic simplification does not carry over to the external entropy flow Equation ([Disp-formula FD4a-entropy-22-01252])
(69)S˙e(t)=−N2∑jk∑nP¯n¯j(t)(Pj(t)wjk−Pk(t)wkj)lnnjwjk(nk+1)wkj=−N2∑jk(Pj(t)wjk−Pk(t)wkj)lnwjkwkjm−N2∑jk∑nP¯n¯j(t)(Pj(t)wjk−Pk(t)wkj)lnnjnk+1,
where of the last two terms only the first is the *N*-fold entropy flow of the single particle system ${\dot{S}}_{e}\left(t\right)$, Equation (60b). The reason for the second term is the lack of a cancellation mechanism to absorb the $n_{j}$ and $n_{k}+1$ from the logarithm. Rewriting the second term as
$$\begin{array}{cc} \; & -\frac{N}{2}\sum_{jk}\sum_{n}{\bar{P}}_{{\bar{n}}_{j}}\left(t\right)\left(P_{j}\left(t\right)w_{jk}-P_{k}\left(t\right)w_{kj}\right)\ln\left(\frac{n_{j}}{n_{k}+1}\right) \end{array}$$
(70)$$\begin{array}{cc} = & -\frac{1}{2}\sum_{n}\sum_{jk}\left(P_{n}\left(t\right)n_{j}w_{jk}-P_{n}\left(\frac{P_{k}\left(t\right)n_{j}}{P_{j}\left(t\right)\left(n_{k}+1\right)}\right)\left(n_{k}+1\right)w_{kj}\right)\ln\left(\frac{n_{j}}{n_{k}+1}\right) \end{array}$$
(71)=−∑nP˙n(t)lnNn1,...,nd,
using Equation (Equation 62) where we re-expressed the logarithm as
(72)lnnjnk+1=lnNn1,…,nj−1,…,nk+1,…,nd−lnNn1,…,nd,
shows that the correction term has the same form as the corresponding term in the two-state system, Equation (Equation 57), namely that of a difference of microcanonical (Boltzmann) entropies of the multi-particle states. It vanishes when all $n_{j}$ are either 0 or 1, as expected for d≫N and also at stationarity when ${\dot{P}}_{n}\left(t\right)=0$. In that limit ${\dot{S}}_{e}=-{\dot{S}}_{i}$ when indeed Equation ([Disp-formula FD60a-entropy-22-01252]) gives
(73)$$\lim_{t\to \infty }{\dot{S}}_{i}^{\left(1\right)}\left(t\right)=\frac{1}{2}\sum_{k}\left(P_{j}w_{jk}-P_{k}w_{kj}\right)\ln\left(\frac{w_{jk}}{w_{kj}}\right)\;,$$
with $P_{j}=\lim_{t\to \infty }P_{j}\left(t\right)$. As far as the entropy production ${\dot{S}}_{i}\left(t\right)$ is concerned, we thus recover and generalise the result in Section 3.5 on indistinguishable particles in a two-state system, which produce *N* times the entropy of a single particle. In Section 3.4, it was shown that *N distinguishable* particles have the same entropy production and flow as the sum of the entropy productions of individual particles. In Section 3.5 and Section 3.6, it was shown that the entropy production of *indistinguishable* particles, which require the states to be represented by occupation numbers, show the *N*-fold entropy production of the single particle system, provided suitable initialisation, but asymptotically independent of initialisation, provided the stationary state has a unique distribution. The same does not apply to the entropy flow, which generally acquires additional logarithmic terms accounting for the degeneracy of the occupation number states. The extra terms, however, are bound to vanish at stationarity, when ${\dot{S}}_{e}\left(t\right)=-{\dot{S}}_{i}\left(t\right)$.

### 3.7. Random Walk on a Lattice

In this section, we study a particle on a one-dimensional lattice that hops to the right nearest neighbouring site with rate *r* and to the left with rate *ℓ*, see Figure 8. The case of unequal hopping rates, ℓ≠r, is generally referred to as an asymmetric random walk and can be seen as a minimal model for the directed motion of a molecular motor on a cytoskeletal filament [41]. The position *x* of the particle at time *t*, after N(t) jumps, is
(74)$$x=x_{0}+\sum_{i=1}^{N(t)}\Delta x_{i},$$
where the random hops $\Delta x_{i}$ are independent and identically distributed, and $x_{0}$ is the initial position at time t=0. If *a* is the lattice spacing, the distance increments are $\Delta x_{i}=+a$ with probability r/(ℓ+r) and $\Delta x_{i}=-a$ with probability ℓ/(ℓ+r). The probability distribution of the particle position is
(75)$$P\left(x,t;x_{0}\right)=\sum_{n=0}^{\infty }H\left(n,t\right)P_{n}\left(x;x_{0}\right)\;,$$
where H(n,t) is the probability that by time *t*, the particle has hopped N(t)=n times, and $P_{n}\left(x;x_{0}\right)$ is the probability that the particle is at position *x* after *n* hops starting from $x_{0}$. Since jumping is a Poisson process with rate r+ℓ, the random variable N(t) has a Poisson distribution,
(76)$$H\left(n,t\right)=\frac{{\left(\left(\ell +r\right)t\right)}^{n}}{n!}e^{-(\ell +r)t}\;.$$
On the other hand, the distribution of the position *x* after *n* jumps is the binomial distribution
(77)Pn(x;x0)=nkxrkxℓn−kx(ℓ+r)n,
where $k_{x}=\left(n+\left(x-x_{0}\right)/a\right)/2$ is the number of jumps to the right, $0\leq k_{x}\leq n$ with (Equation 77) implied to vanish if $k_{x}$ is not integer. From Equation (Equation 74), the parity of $(x-x_{0})/a$ and N(t) are identical. Using (Equation 76) and (Equation 77), the probability distribution in (Equation 75) reads
(78)$$\begin{array}{cc} P(x,t;x_{0})= & e^{-(\ell +r)t}{\left(\frac{r}{\ell }\right)}^{\frac{x-x_{0}}{2a}}I\left(\frac{|x-x_{0}|}{a},2t\sqrt{r\ell }\right)\;, \end{array}$$
where I(n,z) is the modified Bessel function of the first kind of n,z∈C, which is defined as [42]
(79)$$I\left(m,z\right)=\sum_{j=0}^{\infty }\frac{1}{j!\Gamma (j+m+1)}{\left(\frac{z}{2}\right)}^{2j+m}\;.$$
The transition probability is then
(80)$$W\left(x\to y;\tau \right)=e^{-(\ell +r)\tau }{\left(\frac{r}{\ell }\right)}^{\frac{y-x}{2a}}I\left(\frac{\left|y-x\right|}{a},2\tau \sqrt{r\ell }\right)\;.$$

Using (Equation 78) and (Equation 80) to calculate the entropy production (4b), we need the following identity for |y−x|/a=|m|≥1,
(81)$$\lim_{\tau \to 0}\frac{1}{\tau }I\left(|m|,2\tau \sqrt{r\ell }\right)=\sqrt{r\ell }\delta_{|m|,1}\;,$$
which follows immediately from the definition of the modified Bessel function, Equation (Equation 79). It indicates that the only transitions that contribute to the entropy production are those where the particle travels a distance equal to the lattice spacing *a*. Then, the entropy production reads
(82)$$\begin{array}{cc} \;{\dot{S}}_{i}\left(t\right)= & \frac{1}{2}\left(r-\ell \right)\ln\left(\frac{r}{\ell }\right)+e^{-(\ell +r)t}\sum_{m=-\infty }^{\infty }{\left(\frac{r}{\ell }\right)}^{\frac{m}{2}}I\left(|m|,2t\sqrt{r\ell }\right)\\ & \times \left[r\ln\left(\frac{I\left(|m|,2t\sqrt{r\ell }\right)}{I\left(|m+1|,2t\sqrt{r\ell }\right)}\right)+\ell \ln\left(\frac{I\left(|m|,2t\sqrt{r\ell }\right)}{I\left(|m-1|,2t\sqrt{r\ell }\right)}\right)\right]\;, \end{array}$$
see Figure 9. The entropy flow is ${\dot{S}}_{e}\left(t\right)=-\left(r-\ell \right)\ln\left(r/\ell \right)$ independent of *t*, which owes its simplicity to the transition rates being independent of the particle’s position. We are not aware of a method to perform the sum in (Equation 82) in closed form and, given that this expression involves terms competing at large times *t*, we cannot calculate the stationary entropy production $\lim_{t\to \infty }{\dot{S}}_{i}\left(t\right)$. If we naively assume that the sum in Equation (Equation 82) converges such that it is suppressed by the exponential $\exp\left(-(r+\ell )t\right)$, then the entropy production ${\dot{S}}_{i}$ appears to converge to $\frac{1}{2}\left(r-\ell \right)\ln\left(r/\ell \right)$. If that were the case, $\dot{S}={\dot{S}}_{i}+{\dot{S}}_{e}$ would converge to a negative constant, while S(t), Equation (Equation 1), which vanishes at t=0 given the initialisation of $P\left(x,t;x_{0}\right)=\delta_{(x-x_{0})/a,0}$, is bound to be strictly positive at all finite *t*. Given that $P(x,t;x_{0})$ does not converge, not much else can be said about S(t) or $\dot{S}$. Numerically, using the GNU Scientific Library [43] implementation of Bessel functions, we find that, asymptotically for large times, Equation (Equation 82) is
(83)$${\dot{S}}_{i}\left(t\right)≃\left(r-\ell \right)\ln\left(\frac{r}{\ell }\right)+\frac{1}{2t}+O\left(t^{-2}\right)$$
if r≠ℓ and ${\dot{S}}_{i}\left(t\right)≃1/\left(2t\right)+O\left(t^{-3}\right)$ if r=ℓ, see Figure 9.

To take the continuum limit a→0 of the probability distribution (Equation 78), we define *v* and *D* such that $r+\ell =2D/a^{2}$ and r−ℓ=v/a. Using the asymptotic expansion of I(m,z) in *m* [42]
(84)$$I\left(m,z\right)∼\frac{\exp\left(z\right)}{\sqrt{2\pi z}}\left(1-\frac{4m^{2}-1}{8z}+\frac{\left(4m^{2}-1\right)\left(4m^{2}-9\right)}{2!{\left(8z\right)}^{2}}-\frac{\left(4m^{2}-1\right)\left(4m^{2}-9\right)\left(4m^{2}-25\right)}{3!{\left(8z\right)}^{3}}+\ldots \right)\;,$$
which is valid for |argz|<π/2, we obtain in fact the Gaussian distribution,
(85)$$\lim_{a\to 0}\frac{1}{a}P\left(\frac{x}{a},t;\frac{x_{0}}{a};r\left(v,D,a\right),\ell \left(v,D,a\right)\right)=\frac{1}{\sqrt{4\pi Dt}}e^{-\frac{{\left(x-x_{0}-vt\right)}^{2}}{4Dt}}\;,$$
which corresponds to the distribution of a drift–diffusive particle, which is studied in Section 3.9. Therefore, all results derived in Section 3.9, apply to the present system in the continuum limit.

### 3.8. Random Walk on a Ring Lattice

In this section, we extend the system in Section 3.7 to a random walk on a ring lattice of length L>2, so that 1≤x≤L, see Figure 10. The probability distribution $P_{L}\left(x,t\right)$ of the particle on the ring follows from the distribution on the one-dimensional lattice P(x,t) in (Equation 78), by mapping all positions x+jL on the one-dimensional lattice to position x∈{1,2,…,L} on the ring with *j* being the winding number irrelevant to the evolution of the walker. Then, the distribution on the ring lattice reads,
(86)$$P_{L}\left(x,t;x_{0}\right)=\sum_{j=-\infty }^{\infty }P\left(x+jL,t;x_{0}\right)\;$$
and similarly for the transition probability $W\left(x\to y,\tau \right)=P_{L}\left(y,\tau ;x\right)$. To calculate the entropy production (4b), each pair of points x,y on the lattice is mapped to a pair of points on the ring.

For L>2, as τ→0 only transitions to distinct, nearest neighbours contribute and the expression for the entropy production simplifies dramatically,
(87)$${\dot{S}}_{i}\left(t\right)=\left(r-\ell \right)\ln\left(\frac{r}{\ell }\right)+\sum_{m=1}^{L/a}P_{L}\left(ma,t;x_{0}\right)\left\{r\ln\left(\frac{P_{L}\left(ma,t;x_{0}\right)}{P_{L}\left(\left(m+1\right)a,t;x_{0}\right)}\right)+\ell \ln\left(\frac{P_{L}\left(ma,t;x_{0}\right)}{P_{L}\left(\left(m-1\right)a,t;x_{0}\right)}\right)\right\}$$
and similarly for
(88)$${\dot{S}}_{e}\left(t\right)=-\left(r-\ell \right)\ln\left(\frac{r}{\ell }\right).$$
While the entropy flow ${\dot{S}}_{e}$ on a ring is thus identical to that of a particle on a one-dimensional lattice, the entropy production ${\dot{S}}_{i}$ on a ring is in principle more complicated, but with a lack of cancellations of $\sqrt{r/\ell }$ in the logarithm as found in Section 3.7 and $P_{L}$ reaching stationarity comes the asymptote
(89)$$\lim_{t\to \infty }{\dot{S}}_{i}\left(t\right)=\left(r-\ell \right)\ln\left(\frac{r}{\ell }\right).$$
This is easily derived from $\lim_{t\to \infty }P_{L}\left(x,t;x_{0}\right)=1/L$ taken into the finite sum of Equation (Equation 87). It follows that $\dot{S}\left(t\right)={\dot{S}}_{i}\left(t\right)+{\dot{S}}_{e}\left(t\right)$ converges to 0 at large *t*, as expected for a convergent stationary distribution.

The case L=2 and the less interesting case L=1 are not covered above, because of the different topology of the phase space of L>2 compared to L=2. The difference can be observed in the different structure of the transition matrices (Equation 34) and (Equation 39). The framework above is based on each site having two outgoing and two incoming rates, 2L in total. However, for L=2 there are only two transitions, which cannot be separated into four to fit the framework above, because even when rates of concurrent transitions between two given states are additive, their entropy production generally is not. The case of L=2 is recovered in the two-state system of Section 3.1 with α=β=r+ℓ, which is at equilibrium in the stationary state.

### 3.9. Driven Brownian Particle

In continuum space, the motion of a freely diffusive particle with diffusion constant *D* and drift *v* is governed by the Langevin equation $\dot{x}=v+\sqrt{2D}\xi \left(t\right)$, where ξ(t) is a Gaussian white noise with zero mean, $\langle \xi (t)\rangle =0$, and covariance $\langle \xi \left(t\right)\xi \left(t^{\prime }\right)\rangle =\delta \left(t-t^{\prime }\right)$, see Figure 11 [44]. The corresponding Fokker–Planck equation for the probability distribution P(x,t) is [45]
(90)$$\partial_{t}P\left(x,t\right)=-v\partial_{x}P\left(x,t\right)+D\partial_{x}^{2}P\left(x,t\right)\;.$$
Assuming the initial condition $P\left(x,0\right)=\delta \left(x-x_{0}\right)$, the solution to the Fokker–Planck equation is the Gaussian distribution
(91)$$P\left(x,t\right)=\frac{1}{\sqrt{4D\pi t}}e^{-\frac{{\left(x-x_{0}-vt\right)}^{2}}{4Dt}}\;,$$
which is also the Green function of the Fokker–Planck Equation (Equation 90). We therefore also have the transition probability density from state x to state y over an interval τ,
(92)$$W\left(x\to y,\tau \right)=\frac{1}{\sqrt{4D\pi \tau }}e^{-\frac{{\left(y-x-v\tau \right)}^{2}}{4D\tau }}\;.$$

Substituting (Equation 91) and (Equation 92) into Equation (Equation 15) for the internal entropy production of a continuous system gives
(93)$${\dot{S}}_{i}=\lim_{\tau \to 0}\frac{1}{\tau }\int \;dx\;\;dy\;\frac{1}{\sqrt{4D\pi t}}e^{-\frac{{\left(x-x_{0}-vt\right)}^{2}}{4Dt}}\frac{1}{\sqrt{4D\pi \tau }}e^{-\frac{{\left(y-x-v\tau \right)}^{2}}{4D\tau }}\left(\frac{{\left(y-x_{0}\right)}^{2}-{\left(x-x_{0}\right)}^{2}}{4Dt}+\frac{(y-x)v}{2D}\right)\;,$$
where the Gaussian integrals can be evaluated in closed form, ${\dot{S}}_{i}\left(t\right)=\lim_{\tau \to 0}\left[1/\left(2t\right)+v^{2}/D+v^{2}\tau /\left(2Dt\right)\right]$. Taking the limit τ→0 then gives the entropy production rate [44,46,47],
(94)$${\dot{S}}_{i}\left(t\right)=\frac{1}{2t}+\frac{v^{2}}{D}\;,$$
see Figure 12. Similarly, following (Equation 16), the entropy flow reads ${\dot{S}}_{e}\left(t\right)=-v^{2}/D$ independent of time *t*. As ${\dot{S}}_{i}\left(t\right)\neq 0$, we see that for finite *t* or v≠0, the system is out of equilibrium with a sustained probability current, so that there is in fact no steady-state distribution. We can verify Equation (Equation 94) for the time-dependent internal entropy production by computing the probability current
(95)$$j\left(x,t\right)=\left(v-D\partial_{x}\right)P\left(x,t\right)=\left(\frac{v}{2}+\frac{\left(x-x_{0}-vt\right)}{4t}\right)\frac{e^{-\frac{{\left(x-x_{0}-vt\right)}^{2}}{4Dt}}}{\sqrt{\pi Dt}}$$
and substituting it together with (Equation 91), into (Equation 29). As expected, the two procedures return identical results. The independence of the transient contribution 1/(2t) to the internal entropy production on the diffusion constant is remarkable although necessary on dimensional grounds, as a consequence of ${\dot{S}}_{i}$ having dimensions of inverse time. The diffusion constant characterising the spatial behaviour of diffusion suggests that it is the temporal, rather than the spatial, features of the process that determine its initial entropy production.

### 3.10. Driven Brownian Particle in a Harmonic Potential

Consider a drift–diffusive particle such as in Section 3.9 that is confined in a harmonic potential $V\left(x\right)=\frac{1}{2}kx^{2}$, where *k* is the potential stiffness, see Figure 13 [48]. The Langevin equation is $\dot{x}=v-kx+\sqrt{2D}\xi \left(t\right)$, where $\langle \xi (t)\rangle =0$ and $\langle \xi \left(t\right)\xi \left(t^{\prime }\right)\rangle =\delta \left(t-t^{\prime }\right)$ and the Fokker–Planck equation for P(x,t) is [45]
(96)$$\partial_{t}P\left(x,t\right)=-\partial_{x}\left(\left(v-kx\right)P\left(x,t\right)\right)+D\partial_{x}^{2}P\left(x,t\right)\;.$$
Assuming the initial condition $P\left(x,0\right)=\delta \left(x-x_{0}\right)$, the solution to the Fokker–Plank equation is the Gaussian distribution
(97)$$P\left(x,t\right)=\sqrt{\frac{k}{2\pi D(1-\exp\left(-2kt\right))}}e^{-\frac{{\left(kx-v-\left(kx_{0}-v\right)\exp\left(-kt\right)\right)}^{2}}{2Dk(1-\exp\left(-2kt\right))}}\;,$$
corresponding to a probability current $j\left(x,t\right)=\left(v-kx-D\partial_{x}\right)P\left(x,t\right)$ of the form
(98)$$j\left(x\right)=\frac{\sqrt{\frac{k}{2\pi D(1-e^{-2kt})}}e^{-kt}\left(v\left(1-e^{-kt}\right)-k\left(x_{0}-xe^{-kt}\right)\right)}{1-e^{-2kt}}e^{-\frac{(v\left(1-e^{-kt}\right)-k{\left(x-x_{0}e^{-kt}\right)}^{2}}{2Dk(1-e^{-2kt})}}\;.$$
The transition probability density within τ is then also of Gaussian form, namely
(99)$$W\left(x\to y,\tau \right)=\sqrt{\frac{k}{2\pi D(1-\exp\left(-2k\tau \right))}}e^{-\frac{{\left(ky-v-\left(kx-v\right)\exp\left(-k\tau \right)\right)}^{2}}{2Dk(1-\exp\left(-2k\tau \right))}}\;.$$

Using (Equation 97) and (Equation 99) in (Equation 15) gives the entropy production rate
(100)$${\dot{S}}_{i}=\left(\frac{{\left(v-kx_{0}\right)}^{2}}{D}-k\right)e^{-2kt}+\frac{k\exp\left(-2kt\right)}{1-\exp\left(-2kt\right)}\;,$$
see Figure 12, and in (Equation 16) the external entropy flow
(101)$${\dot{S}}_{e}=-\left(\frac{{\left(v-kx_{0}\right)}^{2}}{D}-k\right)e^{-2kt}\;.$$
In the limit t→∞, the system will reach equilibrium as P(x,t) in Equation (Equation 97) converges to the Boltzmann distribution $\sqrt{\frac{k}{2\pi D}}\exp\left(-\frac{{\left(kx-v\right)}^{2}}{2Dk}\right)$ of the effective potential $\frac{1}{2}kx^{2}-vx$ at temperature *D*. This is consistent with (Equation 100) and (Equation 101), since $\lim_{t\to \infty }{\dot{S}}_{i}\left(t\right)=\lim_{t\to \infty }{\dot{S}}_{e}\left(t\right)=0$. Similarly to drift diffusion on the real line, Equation (Equation 94), there is a transient contribution to the entropy production that is independent of the diffusion constant *D* but does now depend on the stiffness *k*, which has dimensions of inverse time, through the rescaled time kt.

### 3.11. Driven Brownian Particle on a Ring with Potential

Consider a drift–diffusive particle on a ring x∈[0,L) in a smooth potential V(x), Figure 14, initialised at position $x_{0}$. The Langevin equation of the particle is [49,50,51] $\dot{x}=v-\partial_{x}V\left(x\right)+\sqrt{2D}\xi \left(t\right)$, where ξ(t) is Gaussian white noise. The Fokker–Planck equation is then
(102)$$\partial_{t}P\left(x,t;x_{0}\right)=-\partial_{x}\left(\left(v-V^{\prime }\left(x\right)\right)P\left(x,t;x_{0}\right)\right)+D\partial_{x}^{2}P\left(x,t;x_{0}\right)$$
with $V^{\prime }\left(x\right)=\frac{d}{dx}V\left(x\right)$ and boundary condition $P^{\left(n\right)}\left(0,t;x_{0}\right)=P^{\left(n\right)}\left(L,t;x_{0}\right)$ for for all n≥0 derivatives and t≥0. At stationarity, in the limit t→∞, where $\partial_{t}P\left(x,t;x_{0}\right)=0$, the solution to the Fokker–Planck Equation (Equation 102) is [29,51,52]
(103)$$P_{s}\left(x\right)=\lim_{t\to \infty }P\left(x,t\right)=Ze^{-\frac{V(x)-vx}{D}}\int_{x}^{x+L}\;dy\;e^{\frac{V(y)-vy}{D}}\;,$$
where Z is the normalisation constant. The corresponding steady-state probability current $j=\left(v-\partial_{x}V\right)P_{s}-D\partial_{x}P_{s}$ is independent of *x* by continuity, $0=\partial_{t}P=-\partial_{x}j$, and reads [45]
(104)$$\begin{array}{cc} j & =Z\left(e^{-\frac{vL}{D}}-1\right). \end{array}$$

In order to calculate the entropy production according to (Equation 15) and (Equation 16) using (Equation 17), we need W(x→y;τ) for small τ. As discussed after Equation (Equation 17), W(x→y;τ) obeys the Fokker–Planck Equation (Equation 102) in the form
(105)$$\partial_{\tau }W\left(x\to y;\tau \right)=-\partial_{y}\left[\left(v-V^{\prime }\left(y\right)\right)W\left(x\to y;\tau \right)\right]+D\partial_{y}^{2}W\left(x\to y;\tau \right)$$
with $\lim_{\tau \to 0}W\left(x\to y;\tau \right)=\delta \left(y-x\right)$, so that
(106)$$\dot{W}\left(x\to y\right)=\lim_{\tau \to 0}\partial_{\tau }W\left(x\to y;\tau \right)=V^{\prime \prime }\left(y\right)\delta \left(y-x\right)-\left(v-V^{\prime }\left(y\right)\right)\delta^{\prime }\left(y-x\right)+D\delta^{\prime \prime }\left(y-x\right)$$
to be evaluated under an integral, where $\delta^{\prime }\left(y-x\right)=\frac{d}{dy}\delta \left(y-x\right)$ will require an integration by parts. As for the logarithmic term, we use [28,45]
(107)$$W\left(x\to y;\tau \right)=\frac{1}{\sqrt{4\pi D\tau }}e^{-\frac{{\left(y-x-\tau \left(v-V^{\prime }\left(x\right)\right)\right)}^{2}}{4D\tau }}\left(1+O(\tau )\right)$$
so that
(108)$$\ln\left(\frac{W(x\to y;\tau )}{W(y\to x;\tau )}\right)=\frac{y-x}{2D}\left(2v-V^{\prime }\left(x\right)-V^{\prime }\left(y\right)\right)+O\left(\tau \right).$$

The entropy flow Equation (Equation 16) in the more convenient version Equation ([Disp-formula FD21a-entropy-22-01252]) can be obtained easily using Equations (Equation 106) and (Equation 108)
(109)$$\begin{array}{cc} {\dot{S}}_{e}\left(t\right) & =-\int_{0}^{L}\;dx\;\;dy\;P\left(x,t\right)\left(V^{\prime \prime }\left(y\right)\delta \left(y-x\right)-\left(v-V^{\prime }\left(y\right)\right)\delta^{\prime }\left(y-x\right)+D\delta^{\prime \prime }\left(y-x\right)\right) \end{array}$$
(110)$$\begin{array}{cc} \; & \;\;\times \frac{y-x}{2D}\left(2v-V^{\prime }\left(x\right)-V^{\prime }\left(y\right)\right) \end{array}$$
(111)$$\begin{array}{cc} \; & =-\int_{0}^{L}\;dx\;P\left(x,t\right)\left(\frac{1}{D}{\left(v-V^{\prime }\left(x\right)\right)}^{2}-V^{\prime \prime }\left(x\right)\right) \end{array}$$
after suitable integration by parts, whereby derivatives of the δ-function are conveniently interpreted as derivatives with respect to *y* to avoid subsequent differentiation of P(x,t). Since δ(y−x)(y−x)=0, the factor (y−x)/(2D) needs to be differentiated for a term to contribute. In the absence of a potential, P(x,t)=1/L at stationarity, so that Equation (Equation 111) simplifies to ${\dot{S}}_{e}\left(t\right)=-v^{2}/D$ and $\lim_{t\to \infty }{\dot{S}}_{i}\left(t\right)=v^{2}/D$, Equation (Equation 94). Using the probability current $j\left(x,t\right)=-D\partial_{x}P\left(x,t\right)+\left(v-V^{\prime }\left(x\right)\right)P\left(x,t\right)$, the entropy flow simplifies further to
(112)$${\dot{S}}_{e}\left(t\right)=-\int_{0}^{L}\;dx\;j\left(x,t\right)\frac{v-V^{\prime }\left(x\right)}{D}$$
so that at stationarity, when the current is spatially uniform, $\lim_{t\to \infty }{\dot{S}}_{e}\left(t\right)=-\lim_{t\to \infty }j\left(x,t\right)vL/D$ as the potential is periodic, entering only via the current.

An equivalent calculation of ${\dot{S}}_{i}$ on the basis of ([Disp-formula FD20a-entropy-22-01252]) gives
(113a)$$\begin{array}{cc} {\dot{S}}_{i}\left(t\right) & =-{\dot{S}}_{e}+\int_{0}^{L}\;dx\;\;dy\;P\left(x,t\right)\left(V^{\prime \prime }\left(y\right)\delta \left(y-x\right)-\left(v-V^{\prime }\left(y\right)\right)\delta^{\prime }\left(y-x\right)+D\delta^{\prime \prime }\left(y-x\right)\right)\ln\left(\frac{P(x,t)}{P(y,t)}\right) \end{array}$$
(113b)$$\begin{array}{cc} \; & =-{\dot{S}}_{e}+\int_{0}^{L}\;dx\;\left\{D\frac{{\left(P^{\prime }\left(x,t\right)\right)}^{2}}{P(x,t)}-P\left(x,t\right)V^{\prime \prime }\left(x\right)\right\} \end{array}$$
(113c)$$\begin{array}{cc} \; & =-{\dot{S}}_{e}-\int_{0}^{L}\;dx\;j\left(x,t\right)\partial_{x}\ln P\left(x,t\right) \end{array}$$
(113d)$$\begin{array}{cc} \; & =\int_{0}^{L}dx\frac{j^{2}\left(x,t\right)}{DP(x,t)}\;, \end{array}$$
with the last line identical to Equation (Equation 13).

By considering the functional derivative δZ/δV(z) in $\int_{0}^{L}dxP\left(x\right)=1$ of Equation (Equation 103), one can show that the stationary current j(x,t) Equation (Equation 104) is extremal for constant V(x), indicating that the magnitude of the stationary entropy flow Equation ([Disp-formula FD113d-entropy-22-01252]) is maximised in a constant potential.

### 3.12. Run-and-Tumble Motion with Diffusion on A ring

Consider the dynamics of a run-and-tumble particle on a ring x∈[0,L) with Langevin equation $\dot{x}=v\left(t\right)+\sqrt{2D}\xi \left(t\right)$, where the drift v(t) is a Poisson process with rate α that alternates the speed of the particle between the constants $v_{1}$ and $v_{2}$, and ξ(t) is Gaussian white noise, Figure 15. Run-and-tumble particles are widely studied as a model of bacterial motion [53]. The drift being $v\left(t\right)=v_{1}$ or $v\left(t\right)=v_{2}$ will be referred to as the mode of the particle being 1 or 2 respectively. Defining $P_{1}\left(x,t\right)$ and $P_{2}\left(x,t\right)$ as the joint probabilities that the particle is at position *x* at time *t* and in mode 1 or 2 respectively, the coupled Fokker–Planck equations for $P_{1}$ and $P_{2}$ are
(114a)$$\begin{array}{cc} \partial_{t}P_{1}\left(x,t\right)= & -v_{1}\partial_{x}P_{1}\left(x,t\right)+D\partial_{x}^{2}P_{1}\left(x,t\right)-\alpha \left(P_{1}\left(x,t\right)-P_{2}\left(x,t\right)\right) \end{array}$$
(114b)$$\begin{array}{cc} \partial_{t}P_{2}\left(x,t\right)= & -v_{2}\partial_{x}P_{2}\left(x,t\right)+D\partial_{x}^{2}P_{2}\left(x,t\right)-\alpha \left(P_{2}\left(x,t\right)-P_{1}\left(x,t\right)\right)\; \end{array}$$
whose stationary solution is the uniform distribution $\lim_{t\to \infty }P_{1}\left(x,t\right)=\lim_{t\to \infty }P_{2}\left(x\right)=1/\left(2L\right)$, as is easily verified by direct substitution. The corresponding steady-state probability currents thus read $j_{1}=v_{1}/\left(2L\right)$ and $j_{2}=v_{2}/\left(2L\right)$.

In the following, we denote by the propagator W(x→y,Q→R;τ) the probability density that a particle at position *x* in mode *Q* is found time τ later at position *y* in mode *R*. For Q=R, this propagation is a sum over all even numbers *m* of Poissonian switches, that occur with probability ${\left(\alpha \tau \right)}^{m}\exp\left(-\alpha \tau \right)/m!$, which includes the probability $\exp\left(-\alpha \tau \right)$ of not switching at all over a total of time τ. For Q≠R, the propagation is due to an odd number of switches.

For m=0, the contribution to W(x→y,Q→R;τ) is thus $\exp\left(-\alpha \tau \right)W(x\to y;\tau )$, with W(x→y;τ) of a drift diffusion particle on a ring, Section 3.11, but without potential, approximated at short times τ by the process on the real line, Equation (Equation 92) with drift $v=v_{1}$ or $v=v_{2}$ according to the particle’s mode. For m=1, the contribution is a single convolution over the time $t^{\prime }\in \left[0,\tau \right)$ at which the particle changes mode, most easily done after Fourier transforming. Before presenting this calculation in real space, we argue that any such convolution will result in some approximate Gaussian with an amplitude proportional to $1/\sqrt{\tau }$ multiplied by a term of order ${\left(\alpha \tau \right)}^{m}$. In small τ, therefore only the lowest orders need to be kept, m=0 for Q=R and m=1 for Q≠R.

More concretely,
$$\begin{array}{cc} \; & W(x\to y,1\to 2;\tau ) \end{array}$$
(115)$$\begin{array}{cc} = & \int_{-\infty }^{\infty }dz\int_{0}^{\tau }d\tau^{\prime }\frac{1}{\sqrt{4\pi D\tau^{\prime }}}e^{-\frac{{\left(z-x-v_{1}\tau^{\prime }\right)}^{2}}{4D\tau^{\prime }}}e^{-\alpha \tau^{\prime }}\frac{1}{\sqrt{4\pi D(\tau -\tau^{\prime })}}e^{-\frac{{\left(y-z-v_{2}\left(\tau -\tau^{\prime }\right)\right)}^{2}}{4D(\tau -\tau^{\prime })}}e^{-\alpha (\tau -\tau^{\prime })}+\ldots \end{array}$$
(116)$$\begin{array}{cc} = & \frac{\alpha \exp\left(-\alpha \tau \right)}{2(v_{1}-v_{2})}\left[\mathrm{erf}\left(\frac{x-y+v_{1}\tau }{\sqrt{4D\tau }}\right)-\mathrm{erf}\left(\frac{x-y+v_{2}\tau }{\sqrt{4D\tau }}\right)\right]+\ldots \end{array}$$
which in small τ, when $v_{1,2}\tau /\sqrt{4D\tau }\ll 1$, so that $\mathrm{erf}\left(r+\varepsilon \right)=\mathrm{erf}\left(r\right)+2\varepsilon \;e^{-r^{2}}/\sqrt{\pi }+\ldots$, expands to
(117)$$\begin{array}{cc} W(x\to y,1\to 2;\tau ) & =\frac{\alpha \tau }{\sqrt{4\pi D\tau }}e^{-\frac{{\left(y-x\right)}^{2}}{4D\tau }}\left(1+O(\tau )\right)\\ \; & =W(x\to y,2\to 1;\tau )\;, \end{array}$$
whereas W(x→y,Q→Q;τ), the propagator with an even number of mode switches, is given by Equation (Equation 92) to leading order in τ,
(118)$$W\left(x\to y,Q\to Q;\tau \right)=\frac{1}{\sqrt{4\pi D\tau }}e^{-\frac{{\left(y-x-v_{Q}\tau \right)}^{2}}{4D\tau }-\alpha \tau }\left(1+O(\tau )\right)\;.$$

Much of the calculation of the entropy production follows the procedure in Section 3.9 and Section 3.11 to be detailed further below. To this end, we also need
(119)$$\begin{array}{cc} \lim_{\tau \to 0}\frac{d}{d\tau }W\left(x\to y,1\to 2;\tau \right) & =\dot{W}\left(x\to y,1\to 2\right)=\alpha \delta \left(x-y\right)\\ \; & =\dot{W}\left(x\to y,2\to 1\right)\;. \end{array}$$
As far as processes are concerned that involve a change of particle mode, therefore only the transition rates enter, not diffusion or drift. Given a uniform stationary spatial distribution of particles of any mode, mode changes between two modes cannot result in a sustained probability current, even when the switching rates differ,
(120)$$\left(P_{1}\dot{W}\left(1\to 2\right)-P_{2}\dot{W}\left(2\to 1\right)\right)\ln\left(\frac{P_{1}\dot{W}\left(1\to 2\right)}{P_{2}\dot{W}\left(2\to 1\right)}\right)=0$$
for $P_{1}\dot{W}\left(1\to 2\right)=P_{2}\dot{W}\left(2\to 1\right)$ at stationarity as in the process discussed in Section 3.1. A probability current and thus entropy production can occur when different particle modes result in a different distribution, Section 3.10, or when mode switching between more than two modes results in a current in its own rights, Section 3.2 and Section 3.13.

Since the full time-dependent density is beyond the scope of the present work, we calculate entropy flow and production at stationary on the basis of a natural extension of Equations (4), (Equation 16) and ([Disp-formula FD21a-entropy-22-01252]) to a mixture of discrete and continuous states
(121)=−limt→∞S˙e(t)=limt→∞S˙i(t)=∑Q,R∈{1,2}∫0LdxdyPQ(x,t)W˙(x→y,Q→R)limτ→0lnW(x→y,Q→R;τ)W(y→x,R→Q;τ)
(122)$$\begin{array}{cc} \; & =\frac{v_{1}^{2}+v_{2}^{2}}{2D} \end{array}$$
which immediately follows from Section 3.9 and Section 3.11, as the stationary density is constant, $P_{Q}=P_{R}=1/\left(2L\right)$, and only Q=R contribute, with
(123)$$\lim_{\tau \to 0}\ln\left(\frac{W(x\to y,1\to 2;\tau )}{W(y\to x,2\to 1;\tau )}\right)=0\;.$$

If the drifts are equal in absolute value $|v_{1}\left|=\right|v_{2}|=v$, then we recover the entropy production of a simple drift–diffusive particle, ${\dot{S}}_{i}=v^{2}/D$. This is because we can think of run-and-tumble as a drift–diffusion particle that changes direction instantly. Since changing the direction produces no entropy, the total entropy production rate should be the same as a drift–diffusion particle. The entropy production can alternatively be derived via (Equation 29) by computing ${\dot{S}}_{i}=\int dx\left(j_{1}^{2}/\left(DP_{1}\right)+j_{2}^{2}/\left(DP_{2}\right)\right)$ with the steady-state currents stated above.

### 3.13. Switching Diffusion Process on a Ring

The dynamics of a one-dimensional run-and-tumble particle discussed above can be readily generalised to the so called switching diffusion process [54] by allowing for an extended set $\left\{v_{i}\right\}$ of drift modes i=1,⋯,M, Figure 16. The corresponding Langevin equation for the particle position on a ring x∈[0,L] is almost identical to that of run-and-tumble, namely $\dot{x}=v\left(t\right)+\sqrt{2D}\xi \left(t\right)$, with the exception that the process v(t) is now an *M*-state Markov process. In the general case, a single switching rate α is thus not sufficient and the full transition rate matrix $\alpha_{ij}$ needs to be provided. In this formulation, the run-and-tumble dynamics Section 3.12 correspond to the choice M=2 with symmetric rates $\alpha_{12}=\alpha_{21}=\alpha$. Defining $P_{i}\left(x,t\right)$ as the joint probability that at time *t* the particle is at position *x* and in mode *i*, thereby moving with velocity $v_{i}$, the system (114) of Fokker–Planck equations generalises to
(124)$$\partial_{t}P_{i}\left(x,t\right)=-\partial_{x}\left[\left(v_{i}-D\partial_{x}\right)P_{i}\left(x,t\right)\right]+\sum_{j}P_{j}\left(x,t\right)\alpha_{ji}$$
where the transmutation rates $\alpha_{ij}$ from mode *i* to mode *j* are assumed to be independent of position. To ease notation we use the convention $\alpha_{jj}=-\sum_{i\neq j}\alpha_{ji}$. For a non-vanishing diffusion constant, the stationary solution is uniform for all modes and given by $\lim_{t\to \infty }P_{i}\left(x,t\right)=z_{i}/L$, where $z_{i}$ is the *i*th element of the eigenvector z satisfying $\sum_{j}z_{j}=1$ and the eigenvalue relation $\sum_{j}z_{j}\alpha_{ji}=0$, which we assume to be unique for simplicity.

The calculation of the steady-state entropy production follows very closely that of run-and-tumble presented above. The conditional transition probabilities including up to one transmutation event read to leading order
(125)$$W\left(x\to y,i\to j;\tau \right)=\left\{\begin{array}{cc} \frac{e^{\alpha_{ii}\tau }}{\sqrt{4\pi D\tau }}e^{-\frac{{\left(y-x-v_{i}\tau \right)}^{2}}{4D\tau }}\left(1+O(\tau )\right)\; & \mathrm{for}i=j\\ \frac{\alpha_{ij}}{2(v_{i}-v_{j})}\left[\mathrm{erf}\left(\frac{x-y+v_{i}\tau }{\sqrt{4D\tau }}\right)-\mathrm{erf}\left(\frac{x-y+v_{j}\tau }{\sqrt{4D\tau }}\right)\right]\left(1+O(\tau )\right)\; & \mathrm{for}i\neq j\;, \end{array}\right.$$
so that
(126)$$\lim_{\tau \to 0}\frac{d}{d\tau }W\left(x\to y,i\to j;\tau \right)=\left\{\begin{array}{cc} D\partial_{y}^{2}\delta \left(y-x\right)-v_{i}\partial_{y}\delta \left(y-x\right)+\alpha_{ii}\delta \left(y-x\right) & \mathrm{for}i=j\\ \alpha_{ij}\delta \left(y-x\right) & \mathrm{for}i\neq j\;. \end{array}\right.$$

We could perform the calculation of the entropy production using the procedure of Section 3.9 rather than drawing on the operator for i=j, which, however, is used in the following for convenience, see Section 3.11. Substituting (Equation 125) and (Equation 126) into ([Disp-formula FD20a-entropy-22-01252]) and assuming steady-state densities, we arrive at
(127)$$\begin{array}{cc} \lim_{t\to \infty }{\dot{S}}_{i}\left(t\right)= & -\lim_{t\to \infty }{\dot{S}}_{e}\left(t\right)\\ = & \int_{0}^{L}dxdy\;\sum_{i}\frac{z_{i}}{L}\left(D\partial_{y}^{2}\delta \left(y-x\right)-v_{i}\partial_{y}\delta \left(y-x\right)+\alpha_{ii}\delta \left(y-x\right)\right)\left(y-x\right)\frac{v_{i}}{D}\\ \; & +\int_{0}^{L}dxdy\;\sum_{i,j\neq i}\frac{z_{i}}{L}\alpha_{ij}\delta \left(y-x\right)\ln\left(\frac{\alpha_{ij}}{\alpha_{ji}}\right)\;, \end{array}$$
where we have used Equation (Equation 126) in the operators containing the δ-functions and Equation (Equation 125) in the logarithms. The term $\ln(\alpha_{ij}/\alpha_{ji})$ is obtained by the same expansion as used in Equation (Equation 117), Section 3.12. Both terms contributing to the entropy production above are familiar from previous sections: the first is a sum over the entropy production of *M* drift–diffusion processes with characteristic drift $v_{i}$, Section 3.11 without potential, weighted by the steady-state marginal probability $z_{i}$ for the particle to be in state *i*; the second is the steady-state entropy production of an M-state Markov process with transition rate matrix $\alpha_{ij}$, which reduces to Equation (4) after integration. Carrying out all integrals, we finally have
(128)$$\lim_{t\to \infty }{\dot{S}}_{i}\left(t\right)=\lim_{t\to \infty }-{\dot{S}}_{e}\left(t\right)=\sum_{i}z_{i}\frac{v_{i}^{2}}{D}+\frac{1}{2}\sum_{i,j}\left(z_{i}\alpha_{ij}-z_{j}\alpha_{ji}\right)\ln\left(\frac{\alpha_{ij}}{\alpha_{ji}}\right).$$
Unlike run-and-tumble, Section 3.12, the transmutation process in switching diffusion does in general contribute to the entropy production for M>2, since the stationary state generally does not satisfy global detailed balance. However, contributions to the total entropy production originating from the switching and those from the diffusion parts of the process are effectively independent at steady state, as only the stationary marginal probabilities $z_{i}$ of the switching process feature as weights in the entropy production of the drift–diffusion. Otherwise, the parameters characterising the two processes stay separate in Equation (Equation 128). Further, the drift–diffusion contributions of the form $v_{i}^{2}/D$ are invariant under the time-rescaling $\alpha_{ij}\to T\alpha_{ij}$. This property originates from the steady-state distributions $P_{i}\left(x\right)$ being uniform and would generally disappear in a potential, Section 3.10.

## 4. Discussion and Concluding Remarks

In this work, we calculate the rate of entropy production within Gaspard’s framework [11] from first principles in a collection of paradigmatic processes, encompassing both discrete and continuous degrees of freedom. Based on the Markovian dynamics of each system, where we can, we derive the probability distribution of the particle (or particles) as a function of time P(x,t) from Dirac or Kronecker-δ initial conditions $P\left(x,0\right)=\delta \left(x-x_{0}\right)$, from which the transition probability W(x→y;τ) follows straightforwardly. In some cases, we determine only the stationary density and the (short-time) propagator W(x→y;τ) to leading order in τ. We then use Equation (4) for discrete systems or Equations (20) and (21) for continuous systems to calculate the time-dependent entropy production. We set out to give concrete, exact results in closed form, rather than general expressions that are difficult to evaluate, even when we allowed for general potentials in Section 3.11. In summary, the ingredients that are needed to calculate the entropy production in closed form in the present framework are: (a) the probability (density) P(x,t) to find the system in state *x* ideally as a function of time *t* and (b) the propagator W(x→y;τ), the probability (density) that the system is found at a certain state *y* after some short time τ given an initial state *x*. If the propagator is known for any time τ, it can be used to calculate the probability $P\left(x,t;x_{0}\right)=W\left(x_{0}\to x;t\right)$ for some initial state $x_{0}$. However, this full time dependence is often difficult to obtain. The propagator is further needed in two forms, firstly $\lim_{\tau \to 0}\partial_{\tau }W\left(x\to y;\tau \right)$ when it is most elegantly written as an operator in continuous space, and secondly $\lim_{\tau \to 0}\ln\left(W\left(x\to y;\tau \right)/W\left(y\to x;\tau \right)\right)$.

For completeness, where feasible, we have calculated the probability current j(x,t) in continuous systems at position *x*. The mere presence of such a flow indicates broken time-reversal symmetry and thus non-equilibrium. Our results on the discrete systems (Section 3.1, Section 3.2, Section 3.3, Section 3.4, Section 3.5, Section 3.6, Section 3.7 and Section 3.8) illustrate two important aspects of entropy production. First, the need of a probability flow $P_{A}\dot{W}\left(A\to B\right)-P_{B}\dot{W}\left(B\to A\right)$ between states: in the two-state system Section 3.1 there are no transition rates α and β such that there is a sustained probability flow and therefore, the system inevitably relaxes to equilibrium. However, in the three-state system Section 3.2 the transition rates can be chosen so that there is a perpetual flow (α−β)/3 between any two states and therefore there is entropy production not only during relaxation but also at stationarity. Hence, we can ascertain these as non-equilibrium steady states in the long term limit due to the non-vanishing rate of internal entropy production. Uniformly distributed steady states can be far from equilibrium, as a rigorous analysis on the basis of the microscopic dynamics reveals, although an effective dynamics may suggest otherwise.

Second, we see how the extensivity of entropy production arises in the *N*-particle systems (Section 3.4, Section 3.5 and Section 3.6), independently of whether the particles are distinguishable or not. We therefore conclude that the number of particles in the system must be accounted for when calculating the entropy production, and doing otherwise will not lead to a correct result. This is sometimes overlooked, especially when using effective theories. In the continuous systems (Section 3.9, Section 3.10 and Section 3.11), which involve a drift *v* and a diffusion constant *D*, we always find the contribution $v^{2}/D$ to the entropy production emerging one way or another. Moreover, in the case of drift–diffusion on the real line (Section 3.9) we find that the contribution due to the relaxation of the system 1/(2t) is independent of any of the system parameters.

Finally, we have studied two systems (Section 3.12 and Section 3.13) where the state space has a discrete and a continuous component. The discrete component corresponds to the transmutation between particle species, i.e., their mode of drifting, whereas the continuous component corresponds to the particle motion. We find that both processes, motion and transmutation, contribute to the entropy production rate essentially independently since any term that combines both processes is a higher-order term contribution in τ, and therefore vanishes in the limit τ→0.

This work has applications to the field of active particle systems, where particles are subject to local non-thermal forces. In fact, the systems studied in Section 3.2, Section 3.8, Section 3.9, Section 3.10, Section 3.11, Section 3.12 and Section 3.13 are prominent examples of active systems. We have shown that their entropy production crucially relies on the microscopic dynamics of the system, which are captured by the Fokker–Planck equation (or the master equation for discrete systems) and its solution. However, in interacting many-particle systems, such a description is not available in general. Instead, we may choose to use the Doi–Peliti formalism [55,56,57,58,59,60,61,62,63] to describe the system, since it provides a systematic approach based on the microscopic dynamics and which retains the particle entity. 

## Figures and Tables

**Figure 1 entropy-22-01252-f001:**
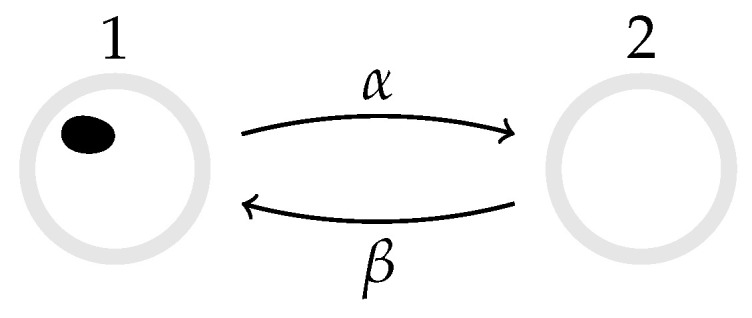
Two-state Markov chain in continuous time. The black blob indicates the current state of the system. Independently of the choice of α and β, this processes settles into an equilibrium steady-state over long timeframes (in the absence of an external time-dependent diving).

**Figure 2 entropy-22-01252-f002:**
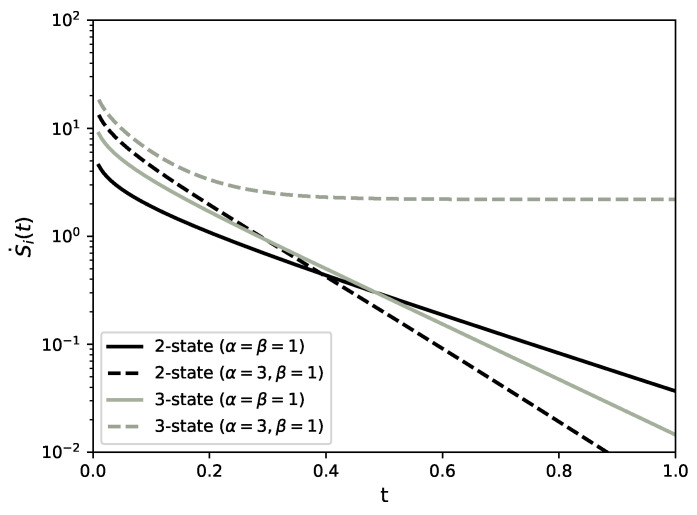
Entropy production of the two- and three-state Markov processes (black and grey lines respectively) discussed in Section 3.1 and Section 3.2 as a function of time. For the two-state system we plot Equation (Equation 37) with p=1 both for the symmetric, α=β (solid lines), and for the asymmetric case, α≠β (dashed lines). In both, the entropy production decays exponentially over long timeframes. For the three-state system, Equation (Equation 41), the asymmetric case displays a finite entropy production rate over long timeframes, consistent with Equation (Equation 42) and the condition that at stationarity ${\dot{S}}_{i}\left(t\right)=-{\dot{S}}_{e}\left(t\right)$.

**Figure 3 entropy-22-01252-f003:**
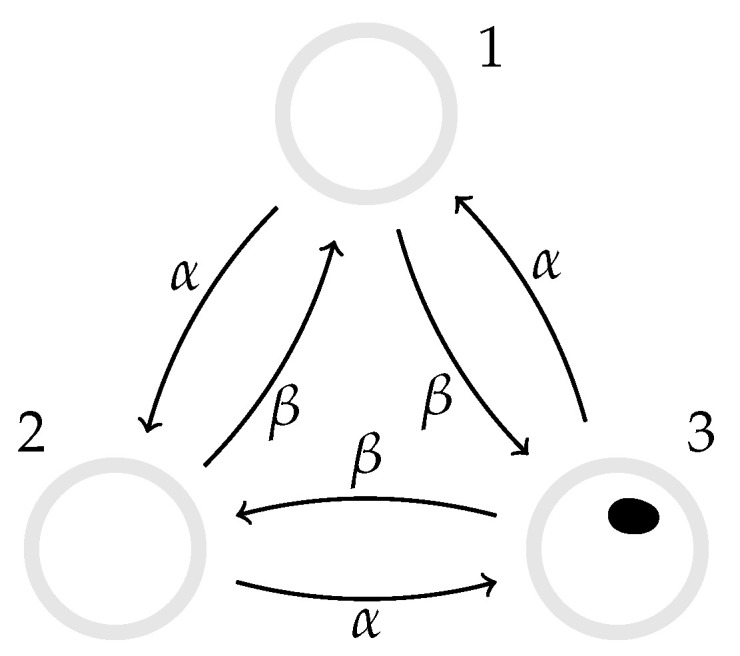
Three-state Markov chain in continuous time. The black blob indicates the current state of the system. Symmetry under cyclic permutation is introduced by imposing identical transition rates α and β for counter-clockwise and clockwise transition, respectively.

**Figure 4 entropy-22-01252-f004:**
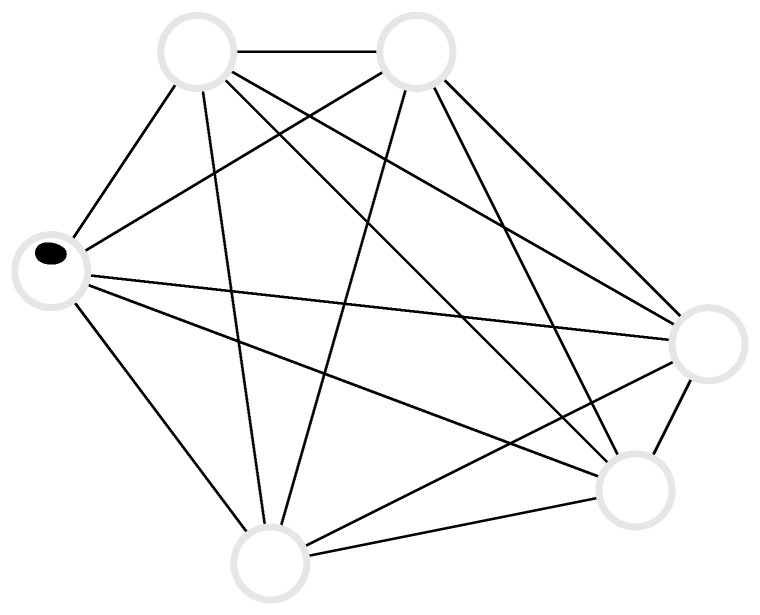
Random walk on a complete graph of *d* nodes (here shown for d=6). The black blob indicates the current state of the system. For uniform transition rates, the symmetry under node relabelling leads to an equilibrium, homogeneous steady-state with $P_{j}=1/d$ for all *j*.

**Figure 5 entropy-22-01252-f005:**
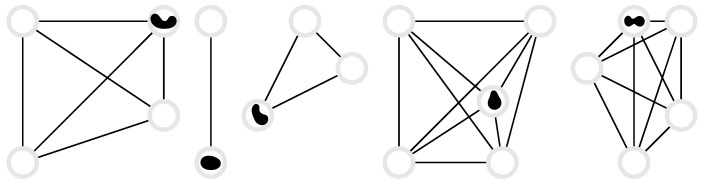
Example of N=5 non-interacting, distinguishable processes with $d_{1}=4$, $d_{2}=2$, $d_{3}=3$, $d_{4}=5$ and $d_{5}=5$. The black blobs indicate the current state of each sub-system.

**Figure 6 entropy-22-01252-f006:**
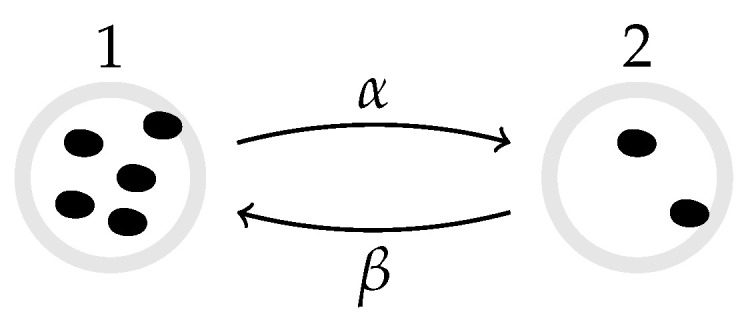
*N* independent, indistinguishable two-state Markov processes in continuous time. The black blobs indicate the current state of the single-particle sub-system. Since processes are indistinguishable, states are fully characterised by the occupation number of either state, if the total number of particles is known.

**Figure 7 entropy-22-01252-f007:**
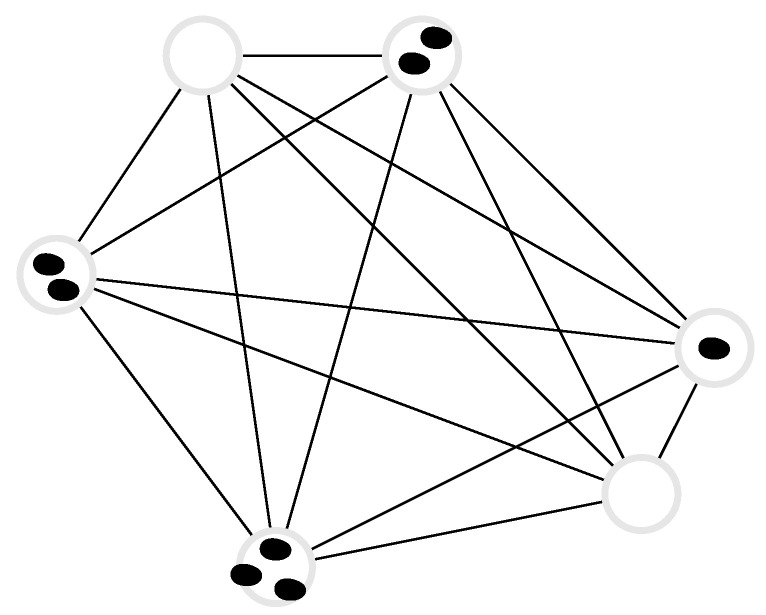
*N* independent, indistinguishable *d*-state Markov processes (here shown for d=6 and N=8) in continuous time. Black blobs indicate the current state of the single-particle sub-systems. Due to indistinguishability, multi-particle states are fully characterised by the occupation number of an arbitrary subset of d−1 states, if the total number of particles is known.

**Figure 8 entropy-22-01252-f008:**
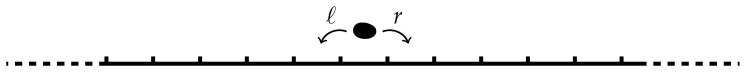
Simple random walk on an infinite, one-dimensional lattice in continuous time. The black blob indicates the current position of the random walker. The left and right hopping rates, labelled *ℓ* and *r* respectively, are assumed to be homogeneous but not equal in general, thus leading to a net drift of the average position.

**Figure 9 entropy-22-01252-f009:**
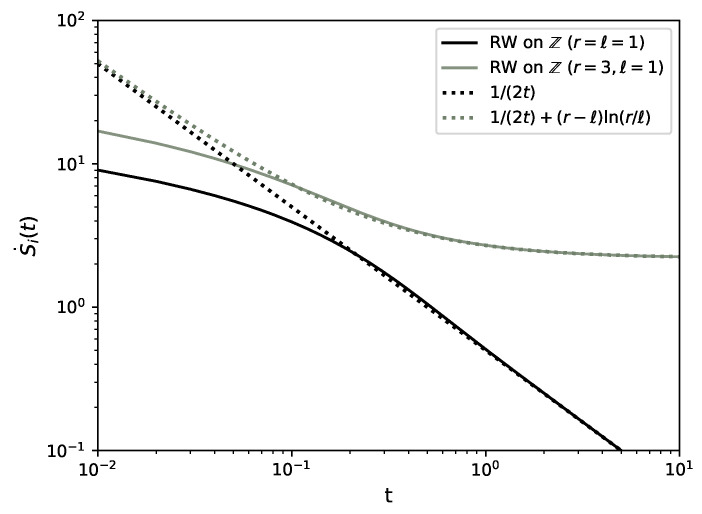
Entropy production of a random walk on a one-dimensional lattice (RW on Z), for symmetric and asymmetric hopping rates, as a function of time, Equation (Equation 82) (solid lines). The asymptotic behaviour at large *t*, Equation (Equation 83) (dotted lines), decays algebraically in the symmetric case (r=ℓ) and converges to a positive constant in the asymmetric case (r≠ℓ).

**Figure 10 entropy-22-01252-f010:**
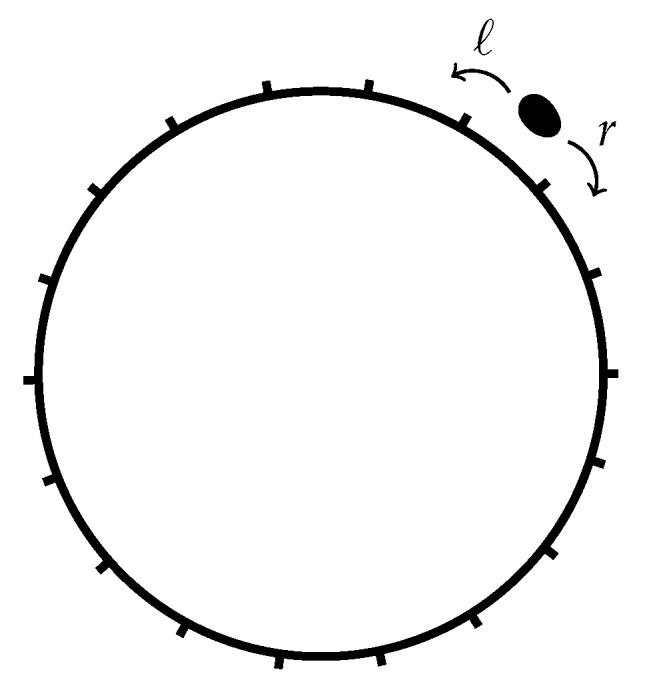
Simple random walk on an periodic, one-dimensional ‘ring’ lattice in continuous time. This model generalises the three-state Markov chain discussed in Section 3.2 to *L* states. The black blob indicates the current position of the random walker. Due to the finiteness of the state space, this process is characterised by a well defined steady-state, which is an equilibrium one for symmetric rates ℓ=r.

**Figure 11 entropy-22-01252-f011:**
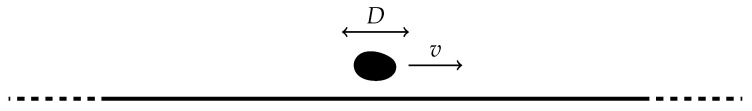
Driven Brownian particle on the real line. The black blob indicates the particle’s current position.

**Figure 12 entropy-22-01252-f012:**
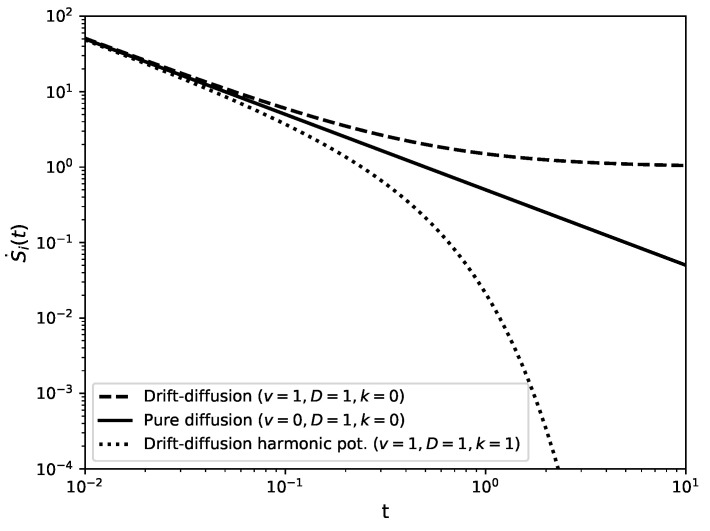
Entropy production of the drift–diffusion process in an external potential, Equation (Equation 100), as a function of time for different parameter combinations. For vanishing potential stiffness, k→0, we recover Equation (Equation 94) for a free drift–diffusion particle. In particular, for v=0 the entropy production decays algebraically, while for v≠0 it converges to the constant value $v^{2}/D$. For k>0, the algebraic decay is suppressed exponentially over long timeframes as the process settles into its equilibrium steady-state.

**Figure 13 entropy-22-01252-f013:**
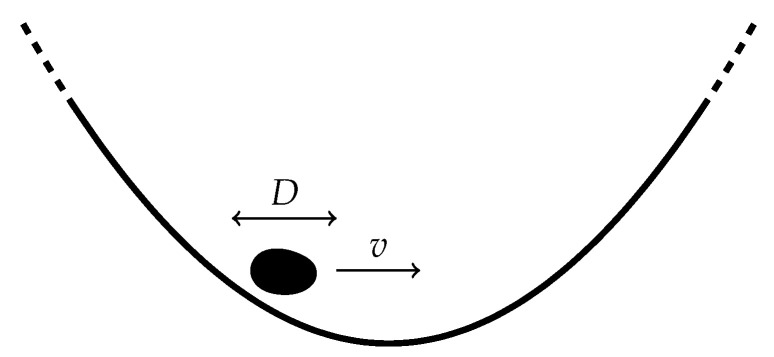
Driven Brownian particle in a harmonic potential. This process reduces to the standard Ornstein–Uhlenbeck process upon rescaling $x\to x^{\prime }+v/k$. The black blob indicates the particle’s current position. The presence of a binding potential implies that the system relaxes to an equilibrium steady-state over long timeframes.

**Figure 14 entropy-22-01252-f014:**
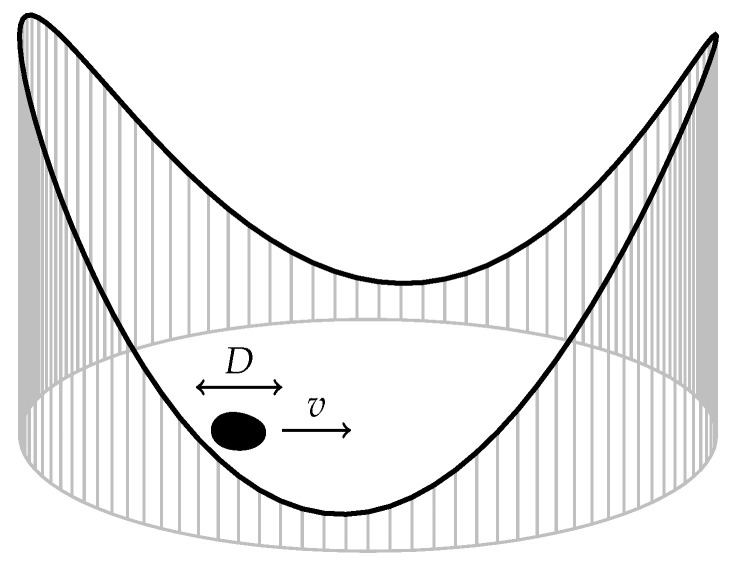
Driven Brownian particle on a ring x∈[0,L) with a periodic potential satisfying V(x)=V(x+L). Any finite diffusion constant D>0 results in a stationary state over long timeframes that is non-equilibrium for v≠0. The black blob indicates the particle’s current position.

**Figure 15 entropy-22-01252-f015:**
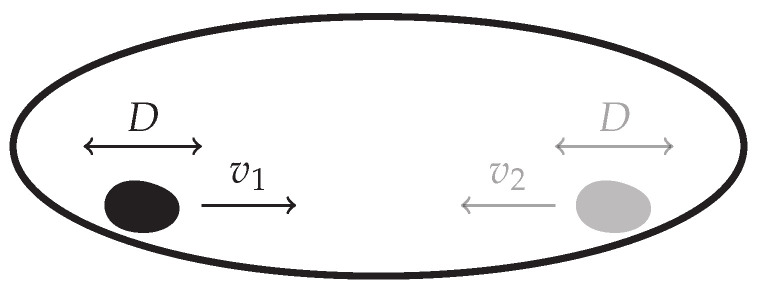
Run-and-tumble motion with diffusion on a ring x∈[0,L). A run-and-tumble particle switches stochastically, in a Poisson process with rate α, between two modes 1 and 2 characterised by an identical diffusion constant *D* but distinct drift velocities $v_{1}$ and $v_{2}$. The two modes are here represented in black and grey, respectively. For arbitrary positive diffusion constant *D* or tumbling rate α with $v_{1}\neq v_{2}$, the steady state is uniform but generally non-equilibrium.

**Figure 16 entropy-22-01252-f016:**
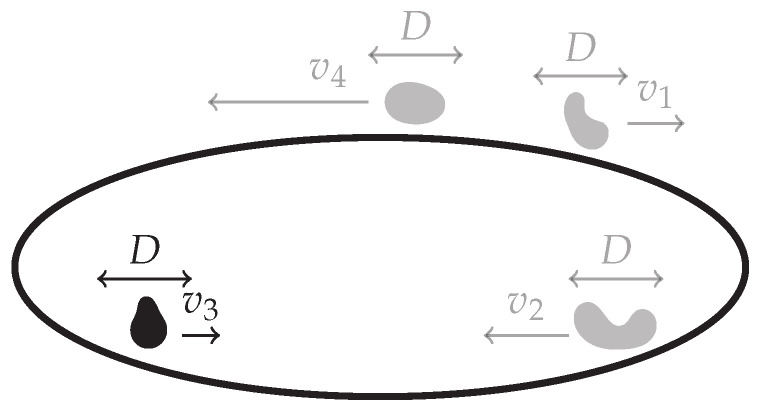
Switching diffusion process on a ring x∈[0,L] in continuous time. A switching diffusion process involves a stochastic switching between *M* modes characterised by an identical diffusion constant *D* but distinct drifts $v_{i}$ (i=1,2,⋯,M). The marginal switching dynamics are characterised as an *M*-state Markov process with transition rates $\alpha_{ij}$ from mode *i* to mode *j*.

**Table 1 entropy-22-01252-t001:** List of particle systems for which we have calculated their entropy production ${\dot{S}}_{i}\left(t\right)$.

Section	System	${\dot{S}}_{i}\left(t\right)$
Section 3.1	Two-state Markov process	(Equation 37)
Section 3.2	Three-state Markov process	(Equation 41)
Section 3.3	Random walk on a complete graph	(Equation 44), (Equation 45)
Section 3.4	*N* independent, distinguishable Markov processes	(Equation 52)
Section 3.5	*N* independent, indistinguishable two-state Markov processes	([Disp-formula FD55b-entropy-22-01252])
Section 3.6	*N* independent, indistinguishable *d*-state processes	(Equation 68)
Section 3.7	Random Walk on a lattice	(Equation 82)
Section 3.8	Random Walk on a ring lattice	(Equation 87), (Equation 89)
Section 3.9	Driven Brownian particle	(Equation 94)
Section 3.10	Driven Brownian particle in a harmonic potential	(Equation 100)
Section 3.11	Driven Brownian particle on a ring with potential	([Disp-formula FD113d-entropy-22-01252])
Section 3.12	Run-and-tumble motion with diffusion on a ring	(Equation 121)
Section 3.13	Switching diffusion process on a ring	(Equation 128)

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
