# Peer review of "Entropy Production in Exactly Solvable Systems"

_entropy, 2020, doi:10.3390/e22111252_

Round 1

Reviewer 1 Report

The authors present a review of analytical results on the entropy
production computed in simple systems. They provide the reader with
definitions and illustrations in discrete and continuous systems,
building on the approach introduced by Gaspard in Ref.[11].

The general presentation is clear and the discussed results are well
organized. This paper could be very useful as introductory material
to researchers approaching the field and as a pedagogical
tool. Therefore, in my opinion, it deserves publication in Entropy.

I only have some minor comments the authors should consider:

1) In the Introduction, due to the review spirit of the paper, some
references to cases where entropy production has been computed in more
complex systems should be added. For instance:
D. Lacoste, A. Lau, and K. Mallick, Physical Review E 78.1 (2008): 011915 (for a model of brownian motor);
A. Puglisi and D. Villamaina, 2009 EPL 88 30004 (for general coupled Langevin equations);
A. Sarracino et al, 2010 EPL 92 34001 (for an application to driven granular systems);
S. Dorosz, and M. Pleimling. Physical Review E 83.3 (2011): 031107 (for transport and reaction-diffusion systems);
G. Gradenigo et al, 2013 J. Phys. A: Math. Theor. 46 335002 (for a stochastic Maxwell–Lorentz particle model);
F. Corberi and A. Sarracino, Entropy 2019, 21(3), 312 (for a recent review with other analytical results);
L. Caprini, et al. Journal of Statistical Mechanics (2019): 053203 (for a recent study in the context of active particles).

2) The problem presented in section 3.7 related to the stationary
limit in eq.(81) could be discussed in more detail. For instance,
could numerical simulations shed light on the problem?

3) Overall, I suggest the authors to add some figures showing the
functional form of the entropy production in some (non-trivial) cases
where analytical expressions are possible (as a function of time or of
some relevant parameters).

4) Typos:
-- In eq.(2) parentheses in the sum are missing
-- line above eq.(21): are --> is
-- eq.(78) does not appear in the text

Author Response

We would like to thank the reviewer for their useful feedback. We have responded to all reviewers in a unified document. Please see the attachment. 

Reviewer 2 Report

Please see the attached review report. 

Author Response

(The authors gave the same response as above.)

Reviewer 3 Report

The authors present an overall view about entropy production and its derivation.

In detail, the authors review the derivation of the entropy production for diverse systems including discrete and continuous Markov processes.

Then they review known systems for which a closed-formula for entropy production is provided.

This review is well written and clearly useful for both experts in the field as well as for beginners.

I recommend for publication in its present form

Author Response

(The authors gave the same response as above.)
